# Long non-coding RNA *Malat1* fine-tunes bone homeostasis and repair by orchestrating cellular crosstalk and β-catenin-OPG/Jagged1 pathway

Yongli Qin[1,2†], Jumpei Shirakawa[1†], Cheng Xu[1], Ruge Chen[1], Xu Yang[3,4], Courtney Ng[1], Shinichi Nakano[1], Mahmoud Elguindy[1], Zhonghao Deng[1], Kannanganattu V Prasanth[5], Moritz F Eissmann[6‡], Shinichi Nakagawa[7], William M Ricci[8], Baohong Zhao[1,2,9]*

[1]Arthritis and Tissue Degeneration Program and David Z. Rosensweig Genomics Research Center, Hospital for Special Surgery, New York, United States; [2]Department of Medicine, Weill Cornell Medical College, New York, United States; [3]Research Institute, Hospital for Special Surgery, New York, United States; [4]Department of Orthopaedic Surgery, Weill Cornell Medicine, New York, United States; [5]Department of Cell and Developmental Biology, Cancer center at Illinois, University of Illinois at Urbana-Champaign, Urbana, United States; [6]Institute for Tumor Biology and Experimental Therapy, Frankfurt, Germany; [7]RNA Biology Laboratory, Faculty of Pharmaceutical Sciences, Hokkaido University, Sapporo, Japan; [8]Orthopaedic Trauma Service, Hospital for Special Surgery & NewYork-Presbyterian Hospital, NewYork, United States; [9]Graduate Program in Cell and Development Biology, Weill Cornell Graduate School of Medical Sciences, New York, United States

**\*For correspondence:**
zhaob@hss.edu

[†]These authors contributed equally to this work

**Present address:** [‡]Olivia Newton-John Cancer Research Institute, and School of Cancer Medicine, La Trobe University, Heidelberg, Victoria, Australia

## eLife Assessment

This is an **important** and **convincing** dataset shedding new light on a role for Malat1 in osteoblast physiology. The work is of value to areas other than the bone field because it supports a role and mechanism for beta-catenin that is novel and unusual. The findings are significant in that they support the presence of another anabolic pathway in bone that can be productively targeted for therapeutic goals. Revisions further improved the paper and addressed the reviewers' concerns.

**Abstract** The lncRNA *Malat1* was initially believed to be dispensable for physiology due to the lack of observable phenotypes in *Malat1* knockout (KO) mice. However, our study challenges this conclusion. We found that both *Malat1* KO and conditional KO mice in the osteoblast lineage exhibit significant osteoporosis. Mechanistically, *Malat1* acts as an intrinsic regulator in osteoblasts to promote osteogenesis. Interestingly, *Malat1* does not directly affect osteoclastogenesis but inhibits osteoclastogenesis in a non-autonomous manner in vivo via integrating crosstalk between multiple cell types, including osteoblasts, osteoclasts, and chondrocytes. Our findings substantiate the existence of a novel remodeling network in which *Malat1* serves as a central regulator by binding to β-catenin and functioning through the β-catenin-OPG/Jagged1 pathway in osteoblasts and chondrocytes. In pathological conditions, *Malat1* significantly promotes bone regeneration in fracture healing. Bone homeostasis and regeneration are crucial to well-being. Our discoveries establish a previous unrecognized paradigm model of *Malat1* function in the skeletal system, providing novel

mechanistic insights into how a lncRNA integrates cellular crosstalk and molecular networks to fine tune tissue homeostasis, remodeling and repair.

## Introduction

Recent genome-wide transcriptome analyses revealed that over 75% of the human genome is transcribed, among which about 95% are transcripts without coding capacity. Long noncoding RNAs (lncRNAs) emerge as a relatively new category of non-coding RNAs (≥500 nucleotides *Mattick et al., 2023*). It is now widely accepted that lncRNAs can perform diverse regulatory roles in regulation of gene expression. lncRNAs are enriched in the nucleus and/or cytoplasm and possess the ability to interact with versatile biomolecules, including various proteins, chromosomal DNAs and RNAs. Therefore, in contrast to miRNAs, lncRNAs regulate gene expression and function by diverse mechanisms, such as functioning as scaffolds for transcriptional and chromatin-modifying complex assemblies, as enhancers or decoys regulating gene transcription, and as cis-acting or trans-acting regulators involved in gene expression and epigenetic regulation (*Batista and Chang, 2013*; *Quinn and Chang, 2016*; *Smith and Mattick, 2017*; *Rinn and Chang, 2012*). Importantly, lncRNAs are druggable targets, and identification of the functional importance of lncRNAs has unveiled new diagnostic and therapeutic opportunities for human diseases, such as cancer, cardiovascular diseases, and genetic disorders (*Batista and Chang, 2013*; *Cech and Steitz, 2014*; *Schmitt and Chang, 2017*; *Modarresi et al., 2012*; *Shappell, 2008*; *Tham et al., 2015*; *Wheeler et al., 2012*; *Wu et al., 2015*; *Yang et al., 2014*). Understanding the biological importance and clinical relevance of lncRNAs is at the forefront of RNA biology research. Nonetheless, only a small fraction of lncRNAs have well-established identifications. Most lncRNAs lack functional annotations, particularly with genetic evidence in vivo and convincing mechanistic studies.

*Malat1* (metastasis associated lung adenocarcinoma transcript 1) is a highly evolutionarily conserved and abundant nuclear lncRNA. It is a 6.8 kilo-nucleotide RNA Polymerase II transcript from mouse chromosome 19 (8.9 kilo-nucleotide from human chromosome 11). As one of the first discovered lncRNAs, *Malat1* was initially recognized as a gene showing specific upregulation in metastatic non-small-cell lung cancer cells (*Gutschner et al., 2013*). Subsequent studies found a variety of associations between *Malat1* and the growth and metastasis of different cancers, such as lung cancer, hepatocellular carcinoma, and breast cancer (*Yoshimoto et al., 2016*; *Kwok et al., 2018*). In physiological settings, although *Malat1* was reported to be a key component of nuclear speckle and a regulator of alternative pre-mRNA splicing in in vitro studies, *Malat1* knockout (KO) mice surprisingly appeared to have no defects in nuclear speckle assembly or pre-mRNA splicing (*Nakagawa et al., 2012*; *Zhang et al., 2012*; *Eißmann et al., 2012*; *Tripathi et al., 2010*; *Ip and Nakagawa, 2012*), raising questions about the extent to which *Malat1* contributes to these processes in vivo. These unexpected discrepancies of in vitro and in vivo studies also draw increasing attention on the importance and necessity of in vivo studies to reveal lncRNA function. The homeostatic/physiological function of *Malat1* in vivo has continued to be an enigma, because the absence of this abundant lncRNA in mice does not seem to exhibit abnormalities (*Nakagawa et al., 2012*; *Zhang et al., 2012*; *Eißmann et al., 2012*). However, to the best of our knowledge, there are no studies that have comprehensively characterized bone phenotype of *Malat1* KO mice.

Bone is a vital organ, which plays a fundamental role in providing structural support to the body, enabling movement, and protecting many other organs. Bone homeostasis is crucial to the quality of life and overall well-being. Bone homeostasis in adulthood is mainly maintained by an active bone remodeling process, which requires a delicate balance between osteoclast-mediated bone resorption and osteoblast-mediated bone formation. Bone tissues undergo constant remodeling, during which bone resorption and formation are usually coupled to ensure that osteoclast-generated resorption lacunae are filled with new bone produced by osteoblasts. This coordination helps maintain bone homeostasis and also provides a mechanism for adapting the skeleton to environmental changes and repairing bone damage. There exists a variety of crosstalk between two major bone cell types, bone-resorbing osteoclasts and bone-forming osteoblasts, as well as other cells. The intricate cellular crosstalk coordinately couples the activities of different cells to maintain bone homeostasis during remodeling (*Sims and Martin, 2014*; *Raggatt and Partridge, 2010*). In pathological conditions, bone remodeling is often deregulated, which results in unbalanced bone resorption and formation. For

example, excessive osteoclast formation accompanied by extensive bone resorption but with limited bone formation/repair often occurs in rheumatoid arthritis (RA), periodontitis, and osteoporosis. On the other hand, excessive bone formation and/or defective bone resorption result in osteopetrosis or osteosclerosis. Notably, unbalanced activities between osteoclasts and osteoblasts in pathological bone remodeling synergize to aggravate the rate and extent of bone damage (*Goldring et al., 2013*; *Schett and Gravallese, 2012*; *Schett and Sieper, 2009*; *Walsh et al., 2009*). Although the importance of bone remodeling is evident to bone homeostasis and health, the coupling and crosstalk mechanisms are complex and far from well understood.

Given the vital importance of bone remodeling to skeletal health and overall well-being, we took advantage of rigorous genetic approaches using global and conditional *Malat1* KO mice in this study. Contrary to previously held beliefs that *Malat1* had no appreciable phenotype, we uncovered that *Malat1* KO mice exhibit a significant osteoporotic bone phenotype characterized by reduced osteoblastic bone formation and enhanced osteoclastic bone resorption in vivo. Thus, *Malat1* deletion uncoupled the normal bone remodeling process between osteoblasts and osteoclasts. Our data further demonstrate that *Malat1* emerges as a novel regulator impacting multiple cell types, including osteoblasts, osteoclasts, and chondrocytes, through the β-catenin-OPG/Jagged1 pathway. This study discovered an important homeostatic function of *Malat1*, and identified this lncRNA as a previously unrecognized key bone remodeling regulator that controls both bone formation and resorption to maintain bone homeostasis, regeneration, and health.

## Results

### *Malat1* is a novel lncRNA regulator of bone homeostasis and remodeling

We first performed a series of experiments to comprehensively examine bone mass, bone resorption, bone formation, bone cell changes and bone remodeling by microCT (μCT), bone histology and dynamic histomorphometric analysis. Adult mice with *Malat1* deficiency (*Malat1*$^{-/-}$ on C57BL/6 background) do not show growth retardation or macroscopic differences (*Figure 1—figure supplement 1A*), but exhibit significantly reduced bone mass compared with wild type littermate controls (WT; *Figure 1A*). μCT analysis showed markedly decreased trabecular bone volume (BV/TV), number (Tb.N.), bone mineral density (BMD) and connectivity density (Conn-Dens.), and increased trabecular spacing (Tb. Sp) in both male and female *Malat1*$^{-/-}$ mice (*Figure 1B*). Cortical bone appears normal (*Figure 1—figure supplement 1B*). Osteoclastic bone resorption was significantly enhanced in *Malat1*$^{-/-}$ mice indicated by the elevated numbers and surfaces of osteoclasts (*Figure 1C and D*). However, osteoblastic bone formation was not accordingly enhanced but instead was dramatically suppressed by *Malat1* deficiency (*Figure 1E–H*). Bone dynamic histomorphometric analysis by calcein double labeling showed that *Malat1* absence notably inhibited both mineral apposition rate (MAR) and bone formation rate (BFR/BS) in *Malat1*$^{-/-}$ mice (*Figure 1E and F*). Furthermore, osteoblast parameters, such as osteoblast surfaces and numbers, as well as osteoid formed beneath osteoblasts were significantly reduced in *Malat1*$^{-/-}$ mice (*Figure 1G and H*). The serum osteoblastic bone formation marker P1NP was decreased, while osteoclastic bone resorption marker TRAP was increased, in *Malat1*$^{-/-}$ mice (*Figure 1I*). These results indicate that the lack of *Malat1* suppresses osteogenesis and bone formation in mice. Moreover, *Malat1* deficiency uncouples osteoclastic bone resorption and osteoblastic bone formation, leading to markedly reduced bone mass. Our data thus identify *Malat1* as a novel bone remodeling regulator in bone homeostasis.

### *Malat1* acts as a cell-intrinsic regulator in osteoblasts to promote bone formation

Next, we examined the role of *Malat1* in osteoblastic bone formation. We crossed *Malat1*$^{flox/flox}$ mice with the mice expressing Cre under the control of *Osteocalcin (Ocn)* promoter to generate *Malat1* conditional KO mice specifically in osteoblasts (*Malat1*$^{flox/flox}$*OcnCre(+)*; hereafter referred to as *Malat1* cKO$^{Ocn}$). *Malat1* deficiency in osteoblasts significantly decreased trabecular bone volume (BV/TV), number (Tb.N.), thickness (Tb. Th), bone mineral density (BMD) and connectivity density (Conn-Dens.), and increased trabecular spacing (Tb. Sp) in *Malat1* cKO$^{Ocn}$ mice (*Figure 2A and B*). There are no obvious changes in cortical bone phenotype (*Figure 2—figure supplement 1*). Furthermore, mineral

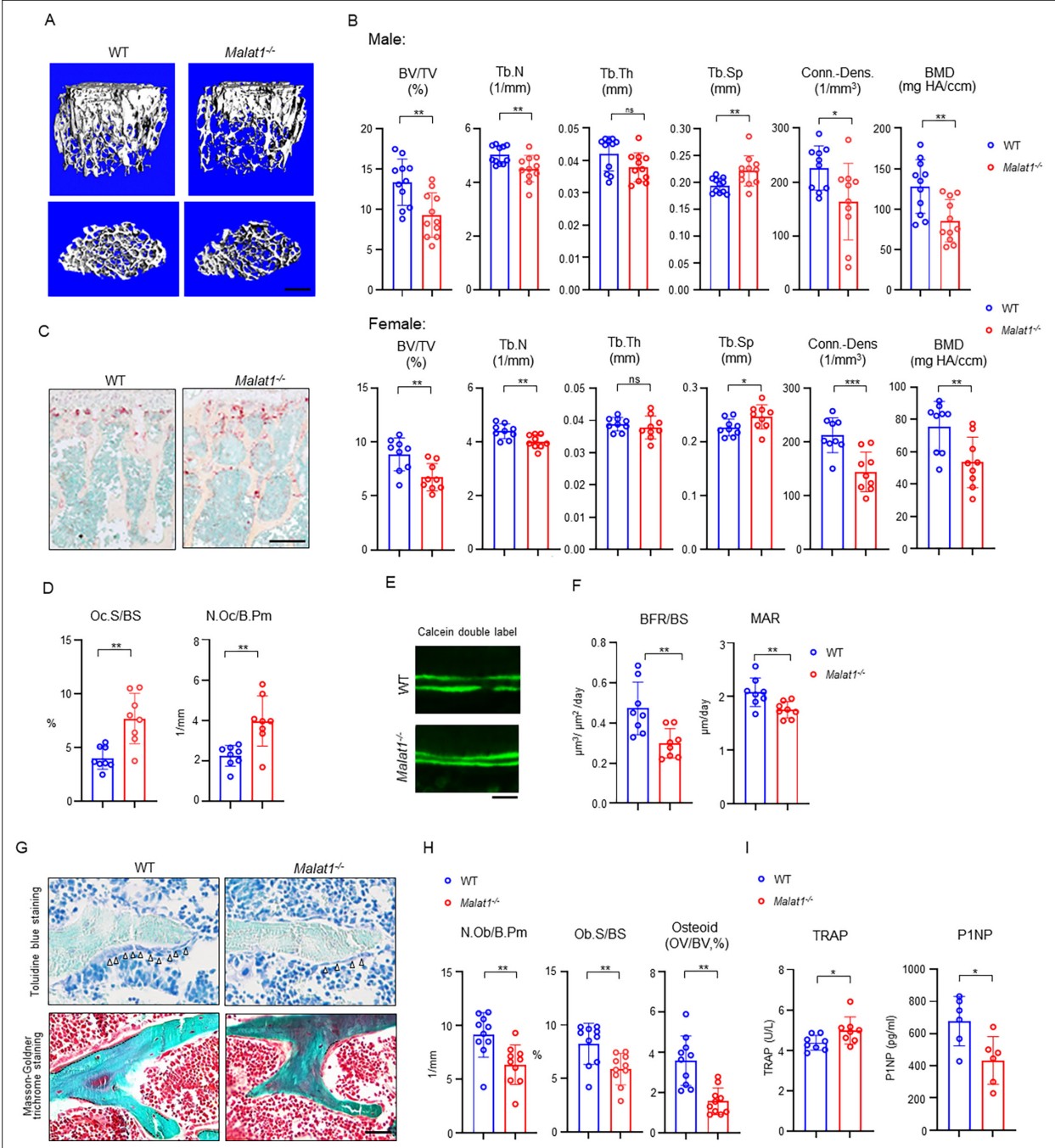

**Figure 1.** *Malat1* deficiency disrupts bone remodeling and results in osteoporosis through reduced osteoblastic bone formation and increased osteoclastic bone resorption. (**A**) µCT images and (**B**) bone morphometric analysis of trabecular bone of the distal femurs isolated from the 12-week-old-male (n=11, upper panel) and female (n=9, lower panel) WT and *Malat1⁻/⁻* littermate mice. BV/TV, bone volume per tissue volume; BMD, bone mineral density; Conn-Dens., connectivity density; Tb.N, trabecular number; Tb.Th, trabecular thickness; Tb.Sp, trabecular separation. (**C**) TRAP staining and (**D**) histomorphometric analysis of histological sections obtained from of 12-week-old male WT and *Malat1⁻/⁻* littermate mice (n = 8/group). Oc.S/BS, osteoclast surface per bone surface; N.Oc/B.Pm, number of osteoclasts per bone perimeter. (**E**) Images of calcein double labelling of the tibia of 12-week-old male WT and *Malat1⁻/⁻* littermate mice. (**F**) Dynamic histomorphometric analysis of mineral apposition rate (MAR) and bone formation rate per bone surface (BFR/BS) after calcein double labeling of the tibiae of WT and *Malat1⁻/⁻* littermate male mice (n = 8/group). (**G**) Representative images of Toluidine blue staining (top) and Masson-Goldner staining (bottom) of femur from 12-week-old-male WT and *Malat1⁻/⁻* littermate mice. For Toluidine blue staining, the bones show green and osteoblasts are indicated by arrow heads. For Masson-Goldner staining, osteoid matrix appears dark orange on the surface of the bone beneath the osteoblasts (indicated by dash lines), osteoblasts are stained orange lining on the bone surface, and bone marrow cells appear red in the photograph. (**H**) Bone morphometric analysis of osteoblast surface per bone surface (Ob.S/BS), osteoblast number per

*Figure 1 continued on next page*

*Figure 1 continued*

bone perimeter (N.Ob/B.Pm) and osteoid matrix volume per bone volume (OV/BV) of the femur of WT and *Malat1*$^{-/-}$ littermate male mice (n = 10/group). **(I)** Serum TRAP and P1NP levels of 12-week-old male mice. **(B, D, F, H, I)** *p < 0.05; **p < 0.01; ns, not statistically significant by Student's t test. Data are mean ± SD. Scale bars: A 400 µm; C 200 µm; E, G 50 µm.

The online version of this article includes the following source data and figure supplement(s) for figure 1:

**Source data 1.** *Malat1* deficiency disrupts bone remodeling and results in osteoporosis through reduced osteoblastic bone formation and increased osteoclastic bone resorption.

**Figure supplement 1.** *Malat1* deficiency does not affect mouse body weight and cortical bone.

**Figure supplement 1—source data 1.** *Malat1* deficiency does not affect mouse body weight and cortical bone.

apposition rate (MAR) and bone formation rate (BFR/BS) were both lower in *Malat1* cKO$^{Ocn}$ mice than the controls (*Figure 2C*). In parallel, *Malat1* deficiency in osteoblasts led to decreased osteoblast numbers and surfaces in vivo (*Figure 2D*). The serum osteoblastic bone formation marker P1NP was decreased in *Malat1* cKO$^{Ocn}$ mice (*Figure 2E*). These results indicate that *Malat1* cKO$^{Ocn}$ mice exhibit osteoporotic phenotype with reduced bone formation, which is consistent with *Malat1*$^{-/-}$ mice. Therefore, *Malat1* is a cell-intrinsic osteogenic regulator that promotes osteoblastic bone formation.

## *Malat1* binds to β-catenin and suppresses its transcriptional activity in osteoblasts

Given these findings, we sought to investigate the mechanisms underlying the regulation of osteoblastic bone formation by *Malat1*. β-catenin is a central transcriptional factor in canonical Wnt signaling pathway, and plays an important role in positively regulating osteoblast differentiation and function (*Zhong et al., 2014*; *Chen and Long, 2013*; *Holmen et al., 2005*; *Kramer et al., 2010*; *Wang et al., 2014*; *Monroe et al., 2012*). Upon stimulation, most notably from canonical Wnt ligands, β-catenin is stabilized and translocates into the nucleus, where it interacts with coactivators to activate target gene transcription. Previous reports observed a link between *Malat1* and β-catenin signaling pathway in cancers *Li et al., 2019*; *Zhang et al., 2018*, but the underlying molecular mechanisms in terms of how *Malat1* interacts with β-catenin and regulates its nuclear retention and transcriptional activity are unclear. In this study, we performed both Chromatin isolation by RNA purification (ChIRP) assay (*Figure 3A, B*, *Figure 3—figure supplement 1*) and RNA immunoprecipitation (RIP) assay (*Figure 3C*) to determine the interaction between *Malat1* and β-catenin. The results obtained by these two approaches clearly show that *Malat1* binds to β-catenin in osteoblasts (*Figure 3A–C*). Next, we asked whether *Malat1* regulates β-catenin nuclear translocation in response to Wnt3a. We did not find that *Malat1* deficiency significantly affects nuclear localization of β-catenin stimulated by Wnt3a (*Figure 3D*, *Figure 3—figure supplement 2*). We then performed Luciferase assay to examine whether *Malat1* modulates the transcriptional activity of β-catenin. The results showed that *Malat1* deficiency significantly reduced the transcriptional activity of β-catenin in response to Wnt3a stimulation (*Figure 3E*). In line with this, the expression levels of β-catenin target genes, such as *Axin2*, *Ccnd1, Lef1 and Myc*, were substantially lower in *Malat1*$^{-/-}$ cells than WT controls (*Figure 3F*). Our findings indicate that *Malat1* is a key regulator of Wnt/β-catenin signaling pathway that is important for osteoblasts. Since *Malat1* is a nuclear lncRNA, these data also suggest that *Malat1* acts as a scaffold to tether β-catenin in nuclei to implement its positive regulation of β-catenin activity.

## *Malat1* inhibits osteoclastogenesis in a non-autonomous manner

As osteoclast formation is significantly increased in *Malat1*$^{-/-}$ mice, we examined whether this osteoclast change is due to an intrinsic role of *Malat1* in these cells. We first investigated in vitro osteoclast differentiation using bone marrow derived macrophages (BMMs) as osteoclast precursors. Surprisingly, there is no significant difference in osteoclast differentiation and mineral resorption between WT and *Malat1*$^{-/-}$ BMM cultures (*Figure 4A and B*), which is inconsistent with the in vivo data (*Figure 1C*). Moreover, the expression of osteoclastogenic transcription factors and osteoclast marker genes was similar between WT and *Malat1*$^{-/-}$ cultures (*Figure 4C and D*). We also generated *Malat1*$^{flox/flox}$ *Lyz2Cre(+)* (*Malat1* cKO$^{Lyz2}$) mice, a myeloid lineage osteoclast specific *Malat1* conditional KO mouse line. The deletion efficiency of *Malat1* in the *Lyz2*-Cre mice is very high (>90%) (*Figure 4E*). However, there is no significant difference in bone mass between *Malat1* cKO$^{Lyz2}$ mice and their littermate control

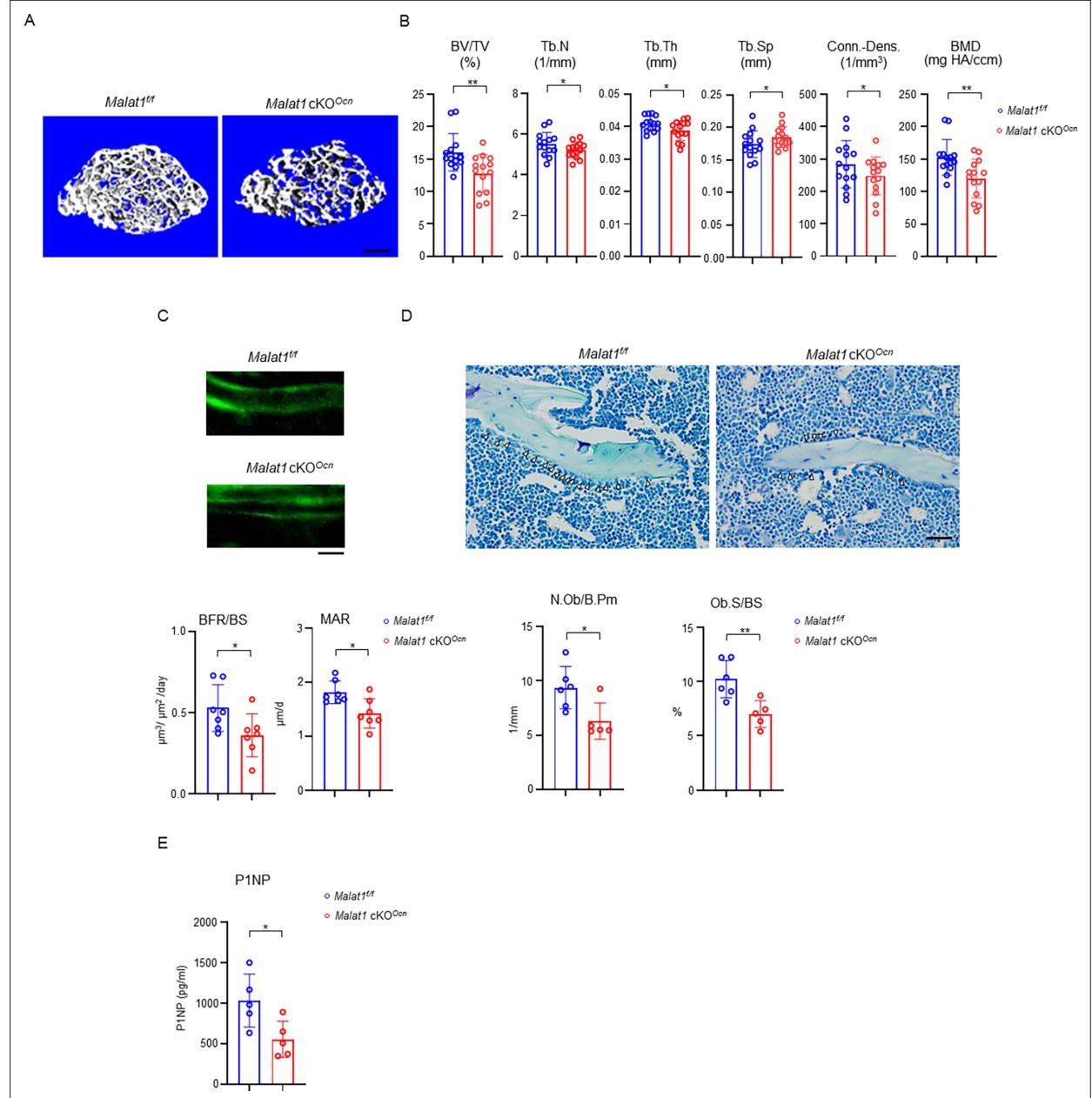

**Figure 2.** Specific deletion of *Malat1* in osteoblasts leads to reduced bone mass and defects in bone formation. (**A**) μCT images and (**B**) bone morphometric analysis of trabecular bone of the distal femurs isolated from the 12-week-old male *Malat1^f/f^* and *Malat1* cKO^Ocn^ littermate mice (n = 14/group). (**C**) Images of calcein double labelling (top) of the tibia of 12-week-old male *Malat1^f/f^* and *Malat1* cKO^Ocn^ littermate mice. Dynamic histomorphometric analysis (bottom) of mineral apposition rate (MAR) and bone formation rate per bone surface (BFR/BS) after calcein double labeling of the tibiae of *Malat1^f/f^* and *Malat1* cKO^Ocn^ littermate male mice (n = 7/group). (**D**) Representative images of Toluidine blue staining (top) of femur from 12-week-old male *Malat1^f/f^* and *Malat1* cKO^Ocn^ littermate mice. For Toluidine blue staining, the bones show green and osteoblasts are indicated by arrow heads. Bone morphometric analysis (bottom) of osteoblast surface per bone surface (Ob.S/BS) and osteoblast number per bone perimeter (N.Ob/B.Pm) of the femur of 12-week-old male *Malat1^f/f^* and *Malat1* cKO^Ocn^ littermate mice. (**E**) Serum P1NP levels of 12-week-old male mice. (**B, C, D, E**) *p < 0.05; **p < 0.01 by Student's t test; ns, not statistically significant. Data are mean ± SD. Scale bars: A 200 μm; C, D 50 μm.

The online version of this article includes the following source data and figure supplement(s) for figure 2:

**Source data 1.** Specific deletion of *Malat1* in osteoblasts leads to reduced bone mass and defects in bone formation.

**Figure supplement 1.** *Malat1* deficiency in *Malat1^ΔOcn^* mice does not affect body weight and cortical bone.

**Figure supplement 1—source data 1.** *Malat1* deficiency in *Malat1* cKO^Ocn^ mice does not affect body weight and cortical bone.

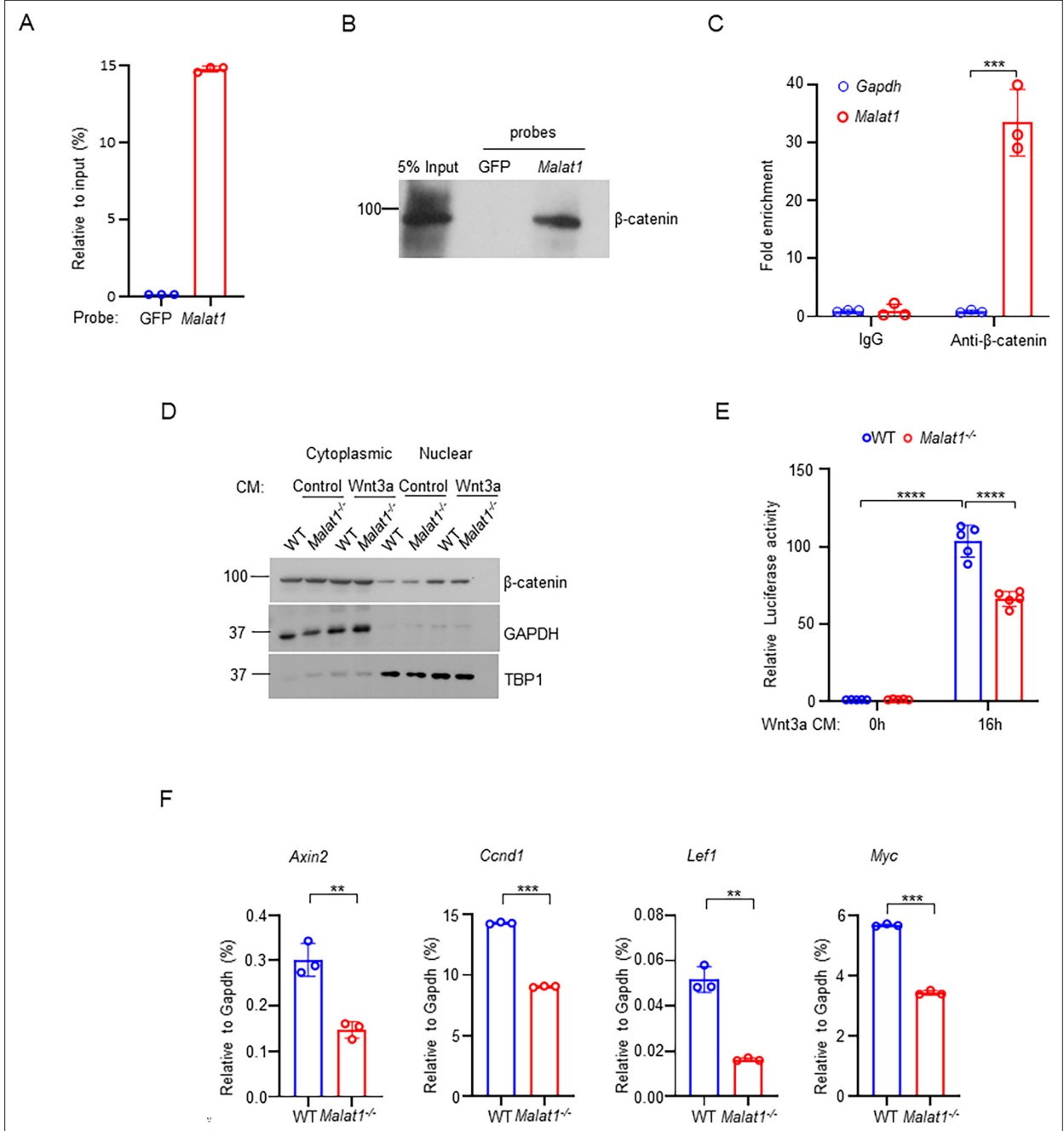

**Figure 3.** *Malat1* binds to β-catenin to positively regulate canonical Wnt/ β-catenin signaling pathway. (**A**) ChIRP analysis of the specificity and efficiency of the *Malat1* probe. Mouse *Malat1* or the control GFP probes were used to pull down endogenous *Malat1* from MC3T3-E1 cells, followed by qPCR quantification of *Malat1*. (**B**) ChIRP analysis of the *Malat1* binding to β-catenin. Mouse *Malat1*-specific probes were used to pull down the endogenous *Malat1* in the MC3T3-E1 cells, followed by immunoblotting with anti-β-catenin antibody. (**C**) RIP assay of β-catenin binding to *Malat1*. Endogenous β-catenin was immunoprecipitated from MC3T3-E1 cells, and the β-catenin-bound *Malat1* was quantitated by qPCR. Rabbit IgG was used as a negative control IP antibody. (**D**) Immunoblot analysis of the nuclear and cytoplasmic localization of β-catenin in calvarial osteoblasts that were serum starved for 16 hr, followed by treatment with 50% Wnt3a- or the control L- conditional medium for 1 hr. TBP1 and GAPDH were measured as loading controls for nuclear and cytoplasmic fractions, respectively. Experiments in a-d were replicated three times. (**E**) Luciferase reporter assay of the Wnt/β-catenin signaling activity measured from the indicated calvarial osteoblasts transfected with the M50 Super 8 x TOPFlash reporter plasmid and pRL-Tk control plasmid for 48 hr, followed by treatment with or without 20% Wnt3a conditional medium for 16 hr (n = 5). (**F**) qPCR analysis of mRNA expression of β-catenin target genes in calvarial osteoblasts in the osteogenic medium (α-MEM with 10% FBS supplemented with 10 mM β-glycerophosphate and 100 ug/ml ascorbic acid) for 7 days (n=3). Data are mean ± SD. (**C, E**) ***p < 0.001; ****p < 0.0001 by two-way ANOVA with Bonferroni's multiple comparisons test. (**F**), **p < 0.01; ***p < 0.001 by Student's t test.

*Figure 3 continued on next page*

*Figure 3 continued*

The online version of this article includes the following source data and figure supplement(s) for figure 3:

**Source data 1.** *Malat1* binds to β-catenin to positively regulate canonical Wnt/ β-catenin signaling pathway.

**Source data 2.** PDF file containing original western blots for *Figure 3B and D*, indicating the relevant bands.

**Source data 3.** Original files for western blot analysis displayed in *Figure 3B and D*.

**Figure supplement 1.** Murine and human *Malat1* probe sequences and their complementary *Malat1* sequences.

**Figure supplement 2.** Immunofluorescence staining of β-catenin.

---

*Lyz2Cre(+)* mice (*Figure 4F and G*). These results demonstrate that *Malat1* expressed in osteoclastic lineage cells does not affect osteoclastogenesis. Therefore, *Malat1* inhibits osteoclastogenesis in a non-autonomous manner in vivo.

## *Malat1* couples osteoblast-osteoclast crosstalk via β-catenin-OPG axis

It is intriguing how *Malat1* regulates osteoclastogenesis. We first looked at osteoclast phenotype in *Malat1* cKO$^{Ocn}$ mice, and found that *Malat1* deletion in osteoblasts significantly enhanced osteoclast numbers and surfaces in vivo (*Figure 5A*). The serum osteoclastic bone resorption marker TRAP was increased in *Malat1* cKO$^{Ocn}$ mice (*Figure 5B*). This finding indicates that *Malat1* affects osteoclast formation through crosstalk between osteoblasts and osteoclasts. To further explore the underlying mechanisms of this crosstalk, we took advantage of a well-established co-culture system (*Zhao et al., 2009*) with primary osteoblasts in a transwell and bone marrow cells on the bottom of the dish. RANKL and M-CSF secreted by osteoblasts induced osteoclast differentiation on the bottom (*Figure 5C*). Interestingly, we found that more osteoclasts formed in the co-cultures with *Malat1$^{-/-}$* osteoblasts than WT osteoblasts (*Figure 5C*). The results of this co-culture system recapitulate the in vivo enhanced osteoclast phenotype, and indicates that certain soluble factors secreted from osteoblasts are highly likely involved in the *Malat1* regulation of osteoclastogenesis. Since we found that *Malat1* binds to β-catenin to regulate its target genes, we primarily searched β-catenin target genes that function as secreted factors. Osteoprotegerin (OPG), encoded by *Tnfrsf11b*, is a β-catenin target (*Boyce et al., 2005*; *Glass et al., 2005*). OPG is a secreted factor that acts as a RANKL decoy receptor to block RANKL activity, thereby suppressing osteoclast formation and bone resorption (*Simonet et al., 1997*). We found that OPG expression was markedly lower in the *Malat1$^{-/-}$* osteoblasts than the WT control cells (*Figure 5D*). RANKL-OPG axis is a typical bone remodeling mediator through impacting osteo-clastogenesis, and RANKL/OPG ratio is usually an indicator for osteoclastogenic/bone resorption potential and status (*Boyce and Xing, 2008*). Our results showed that OPG but not RANKL was down-regulated in *Malat1$^{-/-}$* osteoblasts, leading to a higher RANKL/OPG ratio in *Malat1$^{-/-}$* osteo-blasts than WT controls (*Figure 5E and F*). The elevated RANKL/OPG ratio in *Malat1$^{-/-}$* osteoblasts favors enhanced osteoclastogenesis, which is aligned with the enhanced osteoclast formation and decreased bone mass in *Malat1$^{-/-}$* mice. We next examined the OPG expression in vivo. Recent literature demonstrates that locally produced OPG in bone, but not the serum OPG, is a critical inhibitor of osteoclast formation and bone resorption (*Tsukasaki et al., 2020*). Indeed, the OPG level in serum in *Malat1$^{-/-}$* mice is similar to that in WT controls (*Figure 5G*). However, OPG level in bone marrow was drastically decreased in *Malat1$^{-/-}$* mice compared to WT mice (*Figure 5H*). We further tested the importance of the decreased OPG level in the *Malat1* regulation of osteoclasts. We applied two well-established ex vivo culture systems to closely recapitulate in vivo conditions. In the whole bone marrow cultures with M-CSF and RANKL (*Figure 5I*), *Malat1* deletion in *Malat1$^{-/-}$* bone marrow enabled more osteoclastogenesis than WT bone marrow, which is consistent with the in vivo findings. When recombinant OPG was added to the bone marrow cultures, the osteoclast formation in both WT and *Malat1$^{-/-}$* cultures was inhibited as expected, but the osteoclast formation in *Malat1$^{-/-}$* bone marrow with additional OPG became similar to that in the WT control cultures without additional OPG (*Figure 5I* column 1 vs 4). In another co-culture system (*Figure 5J*), WT or *Malat1$^{-/-}$* osteoblasts were co-cultured with WT bone marrow in the presence of 1,25(OH)$_2$-vitD3 and PGE2. The RANKL secreted from osteoblasts induces osteoclast formation in this co-culture system. Furthermore, because these cultures included the same WT bone marrow cells but different osteoblasts from WT or *Malat1$^{-/-}$* mice, the results directly reflected osteoblastic *Malat1* effects on osteoclast formation. In this co-culture system, we observed a similar phenotype as that in *Figure 5I*. Extra OPG can lead to osteoclast

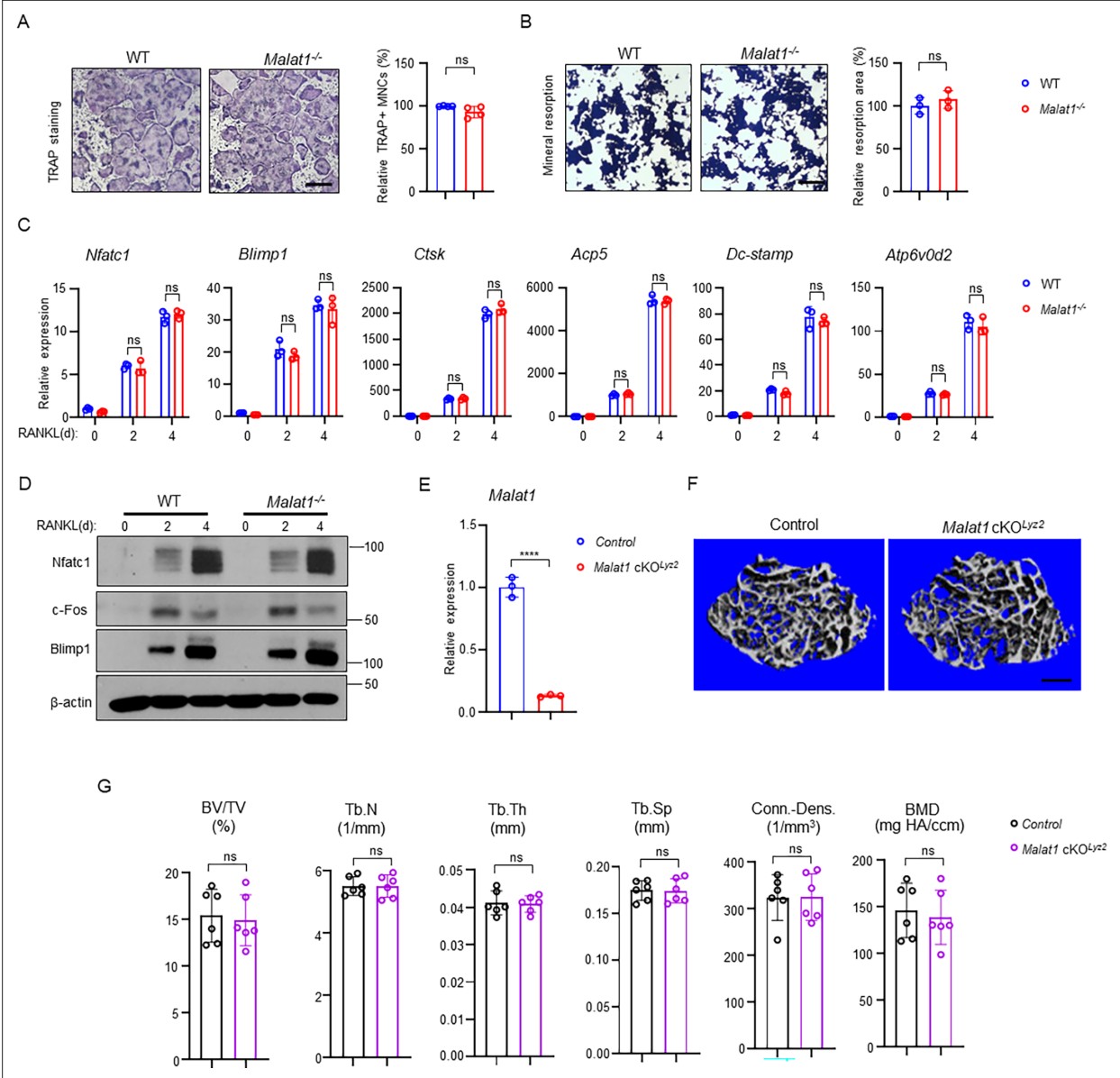

**Figure 4.** *Malat1* is not an intrinsic regulator of osteoclast differentiation. (**A**) Osteoclast differentiation using BMMs obtained from WT and *Malat1⁻/⁻* mice stimulated with RANKL for 3 days. TRAP staining (left panel) was performed and the area of TRAP-positive MNCs (≥3 nuclei/cell) per well relative to the WT control was calculated (right panel). (n = 4/group). (**B**) Von Kossa staining (left) and the resorption area (%) (right) of the osteoclast cultures of WT and *Malat1⁻/⁻* BMMs stimulated with RANKL for 4 days. (n = 3/group). Mineralized area: black; resorption area: white. (**C**) qPCR analysis of mRNA expression of the indicated genes during osteoclastogenesis with or without RANKL for 2 days and 4 days. (**D**) Immunoblot analysis of Nfatc1, Blimp1 and c-Fos expression during osteoclastogenesis with or without RANKL for 2 days and 4 days. β-actin was used as a loading control. (**E–G**) *Malat1* deletion efficiency (**E**) and µCT images (**F**) and bone morphometric analysis (**G**) of trabecular bone of the distal femurs isolated from the indicated 12-week-old male Control and *Malat1* cKO^Lyz2 littermate mice (n = 6/group). Data are mean ± SD. A, B, F ns, not statistically significant by Student's t test; C, by two-way ANOVA with Bonferroni's multiple comparisons test. Scale bars: A,B 100 µm; E 400 µm.

The online version of this article includes the following source data for figure 4:

**Source data 1.** *Malat1* is not an intrinsic regulator of osteoclast differentiation.

**Source data 2.** PDF file containing original western blots for *Figure 4D*, indicating the relevant bands.

**Source data 3.** Original files for western blot analysis displayed in *Figure 4D*.

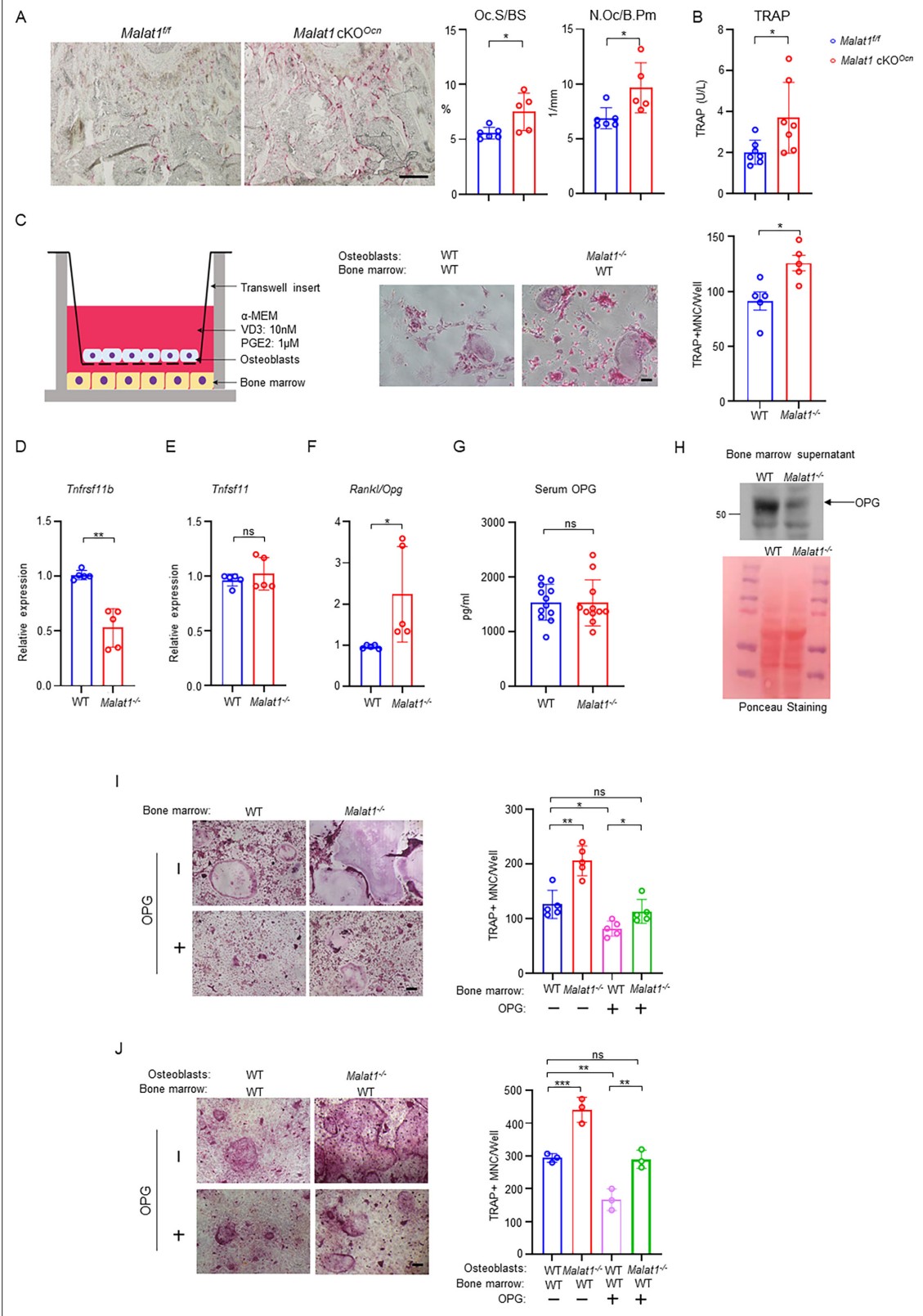

**Figure 5.** *Malat1* promotes OPG expression in osteoblasts to suppress osteoclastogenesis. (**A**) TRAP staining (left) and histomorphometric analysis (right) of histological sections obtained from the metaphysis region of distal femurs from the 12-week-old male *Malat1^f/f^* and *Malat1* cKO^Ocn^ littermate mice. n = 5–6/group. Oc.S/BS, osteoclast surface per bone surface; N.Oc/B.Pm, number of osteoclasts per bone perimeter. (**B**) Serum TRAP levels of 12-week-old male mice. (**C**) A schematic diagram (left) of the co-culture system with primary osteoblasts and bone marrow cells in trans-wells. TRAP

*Figure 5 continued on next page*

*Figure 5 continued*

staining (middle) was performed and the number of TRAP-positive MNCs (≥3 nuclei/cell) per well was calculated (right panel). (n = 5 replicates from two experiments). (**D–E**) qPCR analysis of mRNA expression of *Tnfrsf11b* (encoding OPG) (**D**) and *Tnfsf11* (encoding RANKL) (**E**) in calvarial osteoblasts (n = 5/group). (**F**) The expression ratio of Rankl/Opg in calvarial osteoblasts. (**G**) ELISA analysis of OPG levels in the serum from the 12-week-old male WT and *Malat1*⁻/⁻ mice (n = 11–12/group). (**H**) Immunoblot analysis of OPG expression in the bone marrow supernatant from the 12-week-old male WT and *Malat1*⁻/⁻ mice. Bottom: Ponceau Staining of the gels showing an equivalent amount of total proteins loaded between samples. (**I**) Osteoclast differentiation of WT and *Malat1*⁻/⁻ bone marrows stimulated with RANKL (40 ng/ml) and M-CSF C.M. (1:20) with or without OPG (2.5 ng/ml) for five days. TRAP staining (left panel) was performed and the number of TRAP-positive MNCs (≥3 nuclei/cell) per well was calculated (right panel). TRAP-positive cells appear red in the photographs. n = 5 replicates. (**J**) Osteoclast differentiation of the cocultures of the indicated calvarial osteoblasts and WT bone marrow cells treated with 10 nM of VitD3 and 1 µM of prostaglandinE2 for 6 days in the presence or absence of OPG (1 ng/ml). TRAP staining (left) was performed and the number of TRAP-positive MNCs (≥3 nuclei/cell) per well was calculated (right panel). n = 3 replicates. Data are mean ± SD. A-G, *p < 0.05; **p < 0.01 by Student's t test; I,J *p < 0.05, **p < 0.01, ***p < 0.001 by two-way ANOVA with Bonferroni's multiple comparisons test. ns, not statistically significant. Scale bars: A 200 µm; C,I,J 100 µm.

The online version of this article includes the following source data for figure 5:

**Source data 1.** *Malat1* promotes OPG expression in osteoblasts to suppress osteoclastogenesis.

**Source data 2.** PDF file containing original western blots for **Figure 5H**, indicating the relevant bands.

**Source data 3.** Original files for western blot analysis displayed in **Figure 5H**.

formation in *Malat1*⁻/⁻ osteoblasts and WT bone marrow cocultures comparable to that in WT osteoblasts and WT bone marrow cocultures without extra OPG (**Figure 5J** column 1 vs 4). These results collectively support that the decreased OPG level in *Malat1*-deficient osteoblasts plays a significant role in the excessive osteoclast formation in *Malat1*⁻/⁻ mice (**Figure 1C and D**) and *Malat1* cKO^Ocn mice (**Figure 5A**), as illustrated in Figure 7.

## *Malat1* positively regulates the production of β-catenin targets, OPG and Jagged1, from chondrocytes

As OPG is a key osteoclastic inhibitor, we asked whether other cells, in addition to osteoblasts, in bone marrow could also produce OPG, which we have established is regulated by *Malat1*. We analyzed a bone scRNAseq dataset (GSE128423) (**Baryawno et al., 2019**). Except for osteoblasts as expected, we surprisingly found that chondrocytes express a high level of OPG in the analyzed cells from bone (**Figure 6A-D**, **Figure 6—figure supplement 1**). We then isolated primary chondrocytes from mouse knees, which highly express chondrocyte marker genes, such as *Sox9, Acan, and Col2a1* (**Figure 6E**, **Figure 6—figure supplement 2B**). These cells were also Alcian blue positive (**Figure 6—figure supplement 2A**). We confirmed OPG expression in chondrocytes isolated from mice (**Figure 6G**). Moreover, OPG expression level in *Malat1*⁻/⁻ chondrocytes was approximately 50% less compared to that in WT cells (**Figure 6F and G**). RANKL (encoded by *Tnfsf11*) is nearly undetectable in chondrocytes (**Figure 6F**). The chondrocyte marker genes, including *Sox9, Acan and Col2a1*, were not affected by *Malat1* (**Figure 6E**). These data show that chondrocytes are an important cellular source of OPG highly expressed in bone, which is critically regulated by *Malat1*.

We next examined whether *Malat1* also modulates β-catenin activity in chondrocytes. In addition to OPG, we observed that the expression of other β-catenin target genes, such as *Tcf7, Lef1*, and *Jag1*, was approximately 50% lower in *Malat1*⁻/⁻ chondrocytes than that in WT cells (**Figure 6H**). This suggests that, similar to osteoblasts, *Malat1* positively regulates β-catenin activity in chondrocytes. The decreased expression of Jagged1 in *Malat1*⁻/⁻ chondrocytes drew our attention, as Jagged1 is not only a Notch ligand (**Kopan and Ilagan, 2009**) but also an inhibitor of osteoclastogenesis (**Bai et al., 2008**; **Zhao, 2017**). Following this observation, we evaluated the protein expression levels of Jagged1, and found that *Malat1* deficiency drastically decreased Jagged1 protein expression in chondrocytes (**Figure 6I**). These results collectively indicate that the decreased OPG and Jagged1 production in *Malat1*⁻/⁻ chondrocytes also contributes to the enhanced osteoclast formation and low bone mass in *Malat1*⁻/⁻ mice. Thus, *Malat1* links the activities of chondrocytes and osteoclasts through β-catenin-OPG/Jagged1 axis (**Figure 7**).

## *Malat1* promotes bone regeneration in fracture healing

We next examined whether *Malat1* impacts osteogenesis in pathological conditions. In a femoral midshaft fracture model (**Xu et al., 2018**), we found that *Malat1* deficiency in both *Malat1*⁻/⁻ mice and

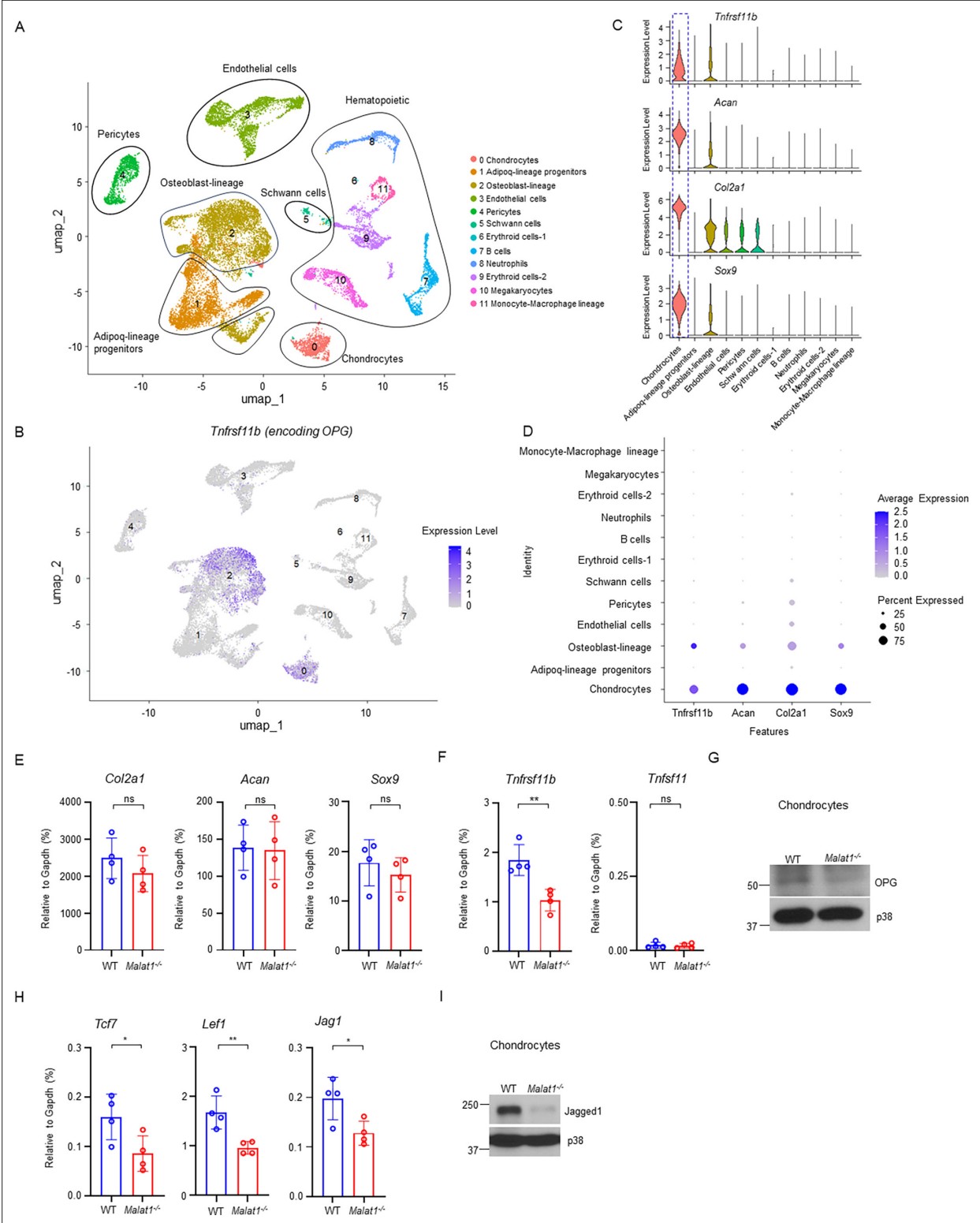

**Figure 6.** *Malat1* enhances OPG and Jagged1 expression in chondrocytes. (**A**) UMAP plot analysis of the bone and bone marrow datasets of scRNAseq based on GSE128423. (**B**) UMAP plot of the expression of *Tnfrsf11b* (encoding OPG) in bone and bone marrow cells. (**C**) Violin plots of the expression of *Tnfrsf11b*, *Acan*, *Col2a1* and *Sox9*. (**D**) Dot plot of the expression of *Tnfrsf11b*, *Acan*, *Col2a1* and *Sox9* across the listed scRNAseq clusters. Cell clusters are listed on y-axis. Features are listed along the x-axis. Dot size reflects the percentage of cells in a cluster expressing each gene. Dot color reflects the scaled average gene expression level as indicated by the legend. (**E, F, H**) qPCR analysis of the indicated genes in primary chondrocytes. n = 4/group.

*Figure 6 continued*

(**G, I**) Immunoblot analysis of OPG and Jagged1 in the chondrocytes isolated from the WT and *Malat1^-/-* mice. Data are mean ± SD. E,F,H, *p < 0.05; **p < 0.01 by Student's t test; ns, not statistically significant.

The online version of this article includes the following source data and figure supplement(s) for figure 6:

**Source data 1.** *Malat1* enhances OPG and Jagged1 expression in chondrocytes.

**Source data 2.** PDF file containing original western blots for *Figure 6G and I*, indicating the relevant bands.

**Source data 3.** Original files for western blot analysis displayed in *Figure 6G and I*.

**Figure supplement 1.** Bioinformatic analysis of the scRNAseq dataset GSE128423.

**Figure supplement 2.** Verification of the cellular characteristics of the primary chondrocytes isolated from mouse knees.

*Malat1* cKO^*Ocn*^ mice significantly disturbed the bone regeneration at 21 days post-fracture, indicated by significantly decreased bone mass in the newly formed callus (*Figure 8*). These results indicate that *Malat1* in osteoblasts plays an important role in enhancing bone regeneration in fracture healing.

## Discussion

*Malat1* stands out as one of the most abundant and evolutionarily conserved long non-coding RNAs (lncRNAs). Initially, the lack of evident defects in *Malat1* knockout (KO) mice led to the assumption that *Malat1* might not be essential for development and physiological processes. However, previous phenotyping studies did not encompass the examination of bone, a vital yet often overlooked organ. Despite its appearance as a hard tissue, bone undergoes continuous remodeling involving various cell types to maintain its mass and function. In this study, we investigated the function of *Malat1* in bone. Our results revealed that *Malat1* acts as a crucial player through the concerted actions of multiple bone cells via a new molecular network in the intricate regulation of bone homeostasis and remodeling under physiological conditions (*Figure 7*). In pathological conditions, such as fracture healing, *Malat1* is an important regulator that promotes bone regeneration. These findings exemplify

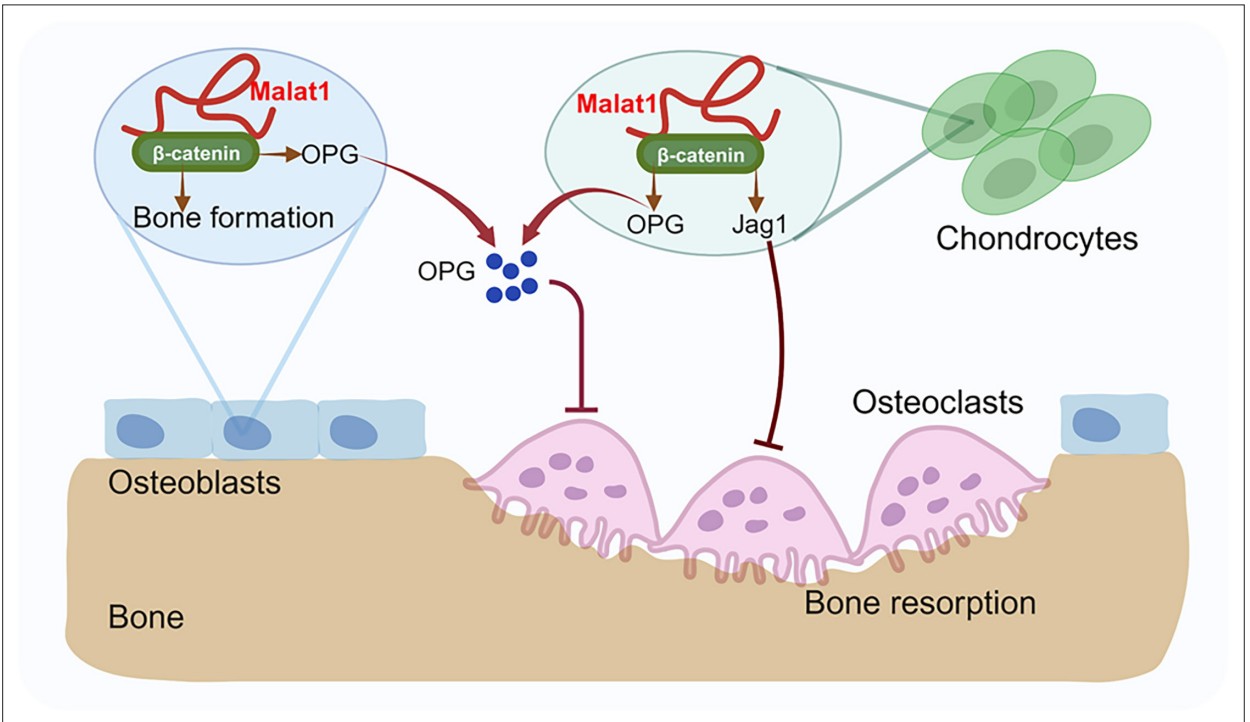

**Figure 7.** A model illustrating a *Malat1*-centered molecular and cellular network in bone remodeling. *Malat1* binds to β-catenin, regulating its transcriptional activity on downstream target genes, such as *Tnfrsf11b* (encoding OPG) and *Jag1* (encoding Jagged1), both of which are osteoclastogenic inhibitors. *Malat1* orchestrates β-catenin to promote intrinsic osteoblastic bone formation while suppressing osteoclastogenesis in a non-autonomous manner through β-catenin target genes OPG and Jagged1, expressed by osteoblasts and chondrocytes.

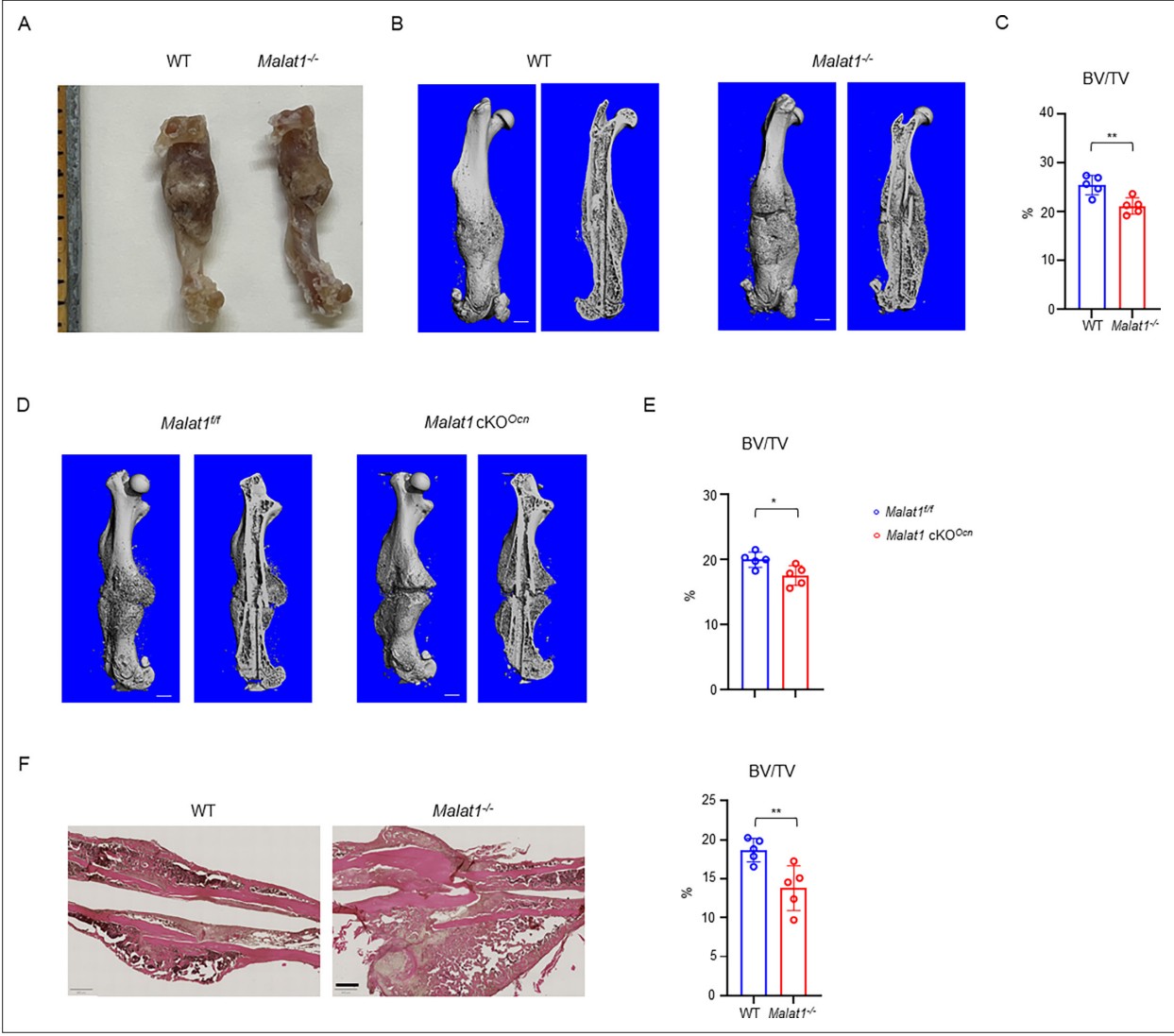

**Figure 8.** *Malat1* enhances bone regeneration in fracture healing. (**A**) Representative photograph of femur fracture callus. (**B, D**) representative μCT images of femurs isolated from the indicated mice at day 21 post-fracture. (**C, E**) μCT analysis of BV/TV in callus area of femurs of the indicated mice at day 21 post-fracture. (**F**) HE staining and histological analysis of the callus areas. n=5/group. Data are mean ± SD. (**C, E, F**) *p < 0.05; **p < 0.01 by Student's t test. Scale bars, B, D 1 mm; F 400 μm.

The online version of this article includes the following source data for figure 8:

**Source data 1.** *Malat1* enhances bone regeneration in fracture healing.

the unprecedented key role of lncRNAs, like *Malat1*, in tissue homeostasis and remodeling, shedding light on their broader significance in orchestrating cellular and molecular networks for maintaining and adapting tissue homeostasis and regeneration.

During the submission of our manuscript, another group (*Zhao et al., 2024*) reported that *Malat1* KO mice show reduced bone mass, which is the only similar phenotype as observed in our study. However, there are many novel findings that are distinct in our study. In addition to global *Malat1* KO mice, we generated both osteoblast and osteoclast specific *Malat1* conditional KO mouse lines. Using the osteoblast-specific conditional KO mice (*Malat1* cKO[Ocn]), we demonstrate that *Malat1* promotes bone formation in osteoblasts. Using the myeloid osteoclast-specific mice (*Malat1* cKO[Lyz2]), we demonstrate that *Malat1* does not directly affect osteoclastogenesis, but suppresses osteoclastogenesis in a non-autonomous manner in vivo by integrating crosstalk between multiple cell types, including osteoblasts, osteoclasts and chondrocytes. We further found that *Malat1* binds to β-catenin and functions through the β-catenin-OPG/Jagged1 pathway in osteoblasts and chondrocytes to enhance bone

formation and suppress bone resorption (*Figure 7*). Moreover, *Malat1* significantly promotes bone regeneration in fracture healing. Taken together, these findings, including the cellular and molecular mechanisms as well as the healing model, are novel and not reported before.

Fine-tuning in biology is essential for maintaining homeostasis (*Lin and Leonard, 2019*; *deLuca and Gommerman, 2012*). Bone must adapt to various environmental changes, ensure optimal function, and respond effectively and precisely to internal and external signals. In this study, we found that *Malat1* is a central player in fine-tuning bone homeostasis and remodeling. *Malat1* promotes osteoblastic bone formation. It also orchestrates the β-catenin pathway to activate the downstream target genes, OPG and Jagged1, which are potent osteoclastogenic inhibitors. Interestingly, *Malat1* does not directly affect osteoclastogenesis in osteoclast cell lineage. The regulation of OPG and Jagged1 by *Malat1* occurs in osteoblasts and chondrocytes. *Malat1* alters the expression of OPG and Jagged1 in these cells, thereby impacting osteoclastogenesis in a non-autonomous manner. Therefore, *Malat1* impacts not only one cell type but fine-tunes a complex cellular and molecular network in the skeletal system. In this network, OPG and Jagged1 are newly identified targets regulated by *Malat1*.

The *Malat1*-β-catenin-OPG/Jagged1 axis identified in our study represents a novel mechanism regulating the crosstalk between chondrocytes and osteoclasts. The exact location of OPG-expressing chondrocytes in cartilage remains unclear. Previous reports using traditional histochemical staining have yielded controversial results, with some indicating OPG expression in articular cartilage and others in growth plates (*Park et al., 2020*; *Chen et al., 2019*; *Kishimoto et al., 2006*; *Silvestrini et al., 2005*). In this study, we do not focus on comparing the sources of OPG from chondrocytes in the growth plate versus articular cartilage. Future studies utilizing advanced technologies would be valuable in elucidating the spatial expression pattern of OPG in cartilage.

Although previous reports observed a link between *Malat1* and β-catenin signaling pathway in cancers (*Li et al., 2019*; *Zhang et al., 2018*), the underlying molecular mechanisms were unclear. Specifically, it is poorly understood how *Malat1* interacts with β-catenin and regulates its nuclear retention and transcriptional activity. Our studies elucidated these questions and uncovered the molecular basis of the *Malat1*- β-catenin pathway. With consideration that lncRNAs can utilize diverse molecular mechanisms, *Malat1* may also bind to additional regulators. Future studies may involve ChIRP assays followed by mass spectrometric analysis to identify additional potential *Malat1*-bound targets.

While studies on lncRNAs using in vivo genetic approaches are increasing, many lncRNA investigations still rely heavily on in vitro methods, such as RNAi techniques for a knockdown in cell cultures. While a knockdown model has the advantage of facilitating high-throughput screens, emerging evidence has revealed significant non-specific or off-target effects in vitro. This has caused subsequent mischaracterizations of lncRNA functions and their respective mechanisms. This is particularly concerning for abundant nuclear lncRNAs, such as *Malat1*. Additional concerns include the transient knockdown effects in vitro versus the stable knockout effects in vivo. The definitive evidence of functionality comes from genetic knockouts, enabling the examination of in vivo function and minimizing the risk of off-target effects. Controversial reports exist between in vitro and in vivo studies regarding *Malat1*'s function in nuclear speckles and cancers (*Nakagawa et al., 2012*; *Zhang et al., 2012*; *Eißmann et al., 2012*; *Tripathi et al., 2010*; *Ip and Nakagawa, 2012*; *Kim et al., 2018*; *Cui et al., 2019*; *Yang et al., 2019*; *Yi et al., 2019*). Recently, some in vitro studies suggested that *Malat1* acts as a miRNA sponge (*Zhou et al., 2021*). However, miRNAs are primarily located in the cytoplasm and lncRNA-miRNA interaction occurs almost exclusively in this cellular compartment. Given *Malat1*'s nuclear localization, it is unlikely to function as an 'miRNA sponge'. At least, the location of the potential interaction between *Malat1* and miRNAs, as well as the *Malat1* copies in extra-nuclear compartments, need to be definitively verified using robust approaches. Other reports on *Malat1*'s functions in bone cells have similar concerns with their in vitro approaches (*Cui et al., 2019*; *Yang et al., 2019*; *Yi et al., 2019*). Overall, the distinct results obtained from in vitro knockdown systems and in vivo Malat1 knockout mouse models provide a clear example of the limitations of in vitro knockdown systems for annotating lncRNA function in vivo. Therefore, discussions in the lncRNA field have emphasized the robustness of approaches, and highlighted the critical need to apply genetic methods aptly to confidently characterize lncRNA functions (*Nakagawa et al., 2012*; *Eißmann et al., 2012*; *Kim et al., 2018*). This is especially relevant in in vivo contexts such as development, metabolism, homeostasis, and tissue remodeling. The *Malat1*[-/-] mice *Nakagawa et al., 2012* used in this study were generated by inserting beta-galactosidase gene followed by a polyadenylation signal immediately downstream

from the transcriptional start site of *Malat1*. This strategy preserves DNA regulatory elements at the *Malat1* locus, thereby avoiding confusion in result interpretation arising from the loss of these elements.

A challenge in lncRNA field is the unclear structures of most lncRNAs. The size of lncRNAs typically spans from 1 kb to over 100 kb. These molecules exhibit intricate secondary and tertiary structures that become increasingly dynamic and complex based on their interactions with various proteins. The structural flexibility of lncRNAs in diverse cellular contexts poses a more significant challenge in deciphering the relationship between their structure and function (*Mattick et al., 2023*). For instance, in a model of estimating lncRNA structures, approximately half of *Malat1* nucleotides were found to adopt various structures, including nearly 200 helices, many pseudoknots, structured tetraloops, internal loops, and intricate intramolecular long-range interactions featuring multiway junctions (*McCown et al., 2019*). To unravel these complexities, breakthrough techniques are required to understand the functions of each and all of the dynamic structures of lncRNAs.

Many skeletal diseases involve defects in bone remodeling, a process that includes multiple cell types, cellular crosstalk, and complex molecular pathways. Treatment efficacy is usually insufficient by targeting only one bone cell type. This is clearly evident in diseases such as RA and periodontitis, in which standard antiresorptive therapies, such as bisphosphonates that inhibit osteoclast activity, are not able to effectively restore bone formation (*Goldring et al., 2013*; *Schett and Gravallese, 2012*; *Schett and Sieper, 2009*; *Walsh et al., 2009*). Therefore, there is a clinical need for new or complementary therapies that can target multiple bone cell types, such as osteoblasts and osteoclasts. These innovative therapeutic strategies go beyond targeting a single cell type, aiming instead to restore healthy bone remodeling and significantly enhance treatment efficacy. *Malat1*, identified in this study, emerges as a crucial fine-tuner that integrates multiple bone cells and a molecular network to maintain bone homeostasis. Thus, *Malat1* and its mediated cellular and molecular mechanisms not only hold significant implications for our understanding of conditions related to bone health but also offer a potential avenue for developing more efficient treatments for bone diseases.

# Materials and methods

**Key resources table**

| Reagent type (species) or resource | Designation | Source or reference | Identifiers | Additional information |
|---|---|---|---|---|
| Genetic reagent (*M. musculus*) | *Malat1*⁻/⁻ | PMID:22718948 | | |
| Genetic reagent (*M. musculus*) | *Malat1*^flox/flox^ | PMID:22858678 | | |
| Genetic reagent (*M. musculus*) | Osteocalcin cre | PMID:12215457 | RRID:IMSR_JAX:019509 | |
| Genetic reagent (*M. musculus*) | *Lyz2* cre | PMID:10621974 | RRID:IMSR_JAX:004781 | |
| Antibody | β-catenin rabbit polyclonal | Cell Signalling Technology | Cat# 9562 RRID:AB_331149 | WB (1:1000) |
| Antibody | Jag1 Rabbit monoclonal | Cell Signalling Technology | Cat# 70109 RRID:AB_331149 | WB (1:1000) |
| Antibody | Nfatc1 Mouse monoclonal | BD Biosciences | Cat# 556602 RRID:AB_331149 | WB (1:1000) |
| Antibody | Blimp1 Rat monoclonal | Santa Cruz Biotechnology | Cat# sc-47732 RRID:AB_628168 | WB (1:1000) |
| Antibody | c-Fos Rabbit polyclonal | Santa Cruz Biotechnology | Cat# sc-52 RRID:AB_2106783 | WB (1:1000) |
| Antibody | OPG/Osteoprotegerin Mouse monoclonal | Santa Cruz Biotechnology | Cat# sc-390518 RRID:AB_2891104 | WB (1:1000) |
| Antibody | p38α Rabbit polyclonal | Santa Cruz Biotechnology | Cat# sc-535 RRID:AB_632138 | WB (1:1000) |

*Continued on next page*

*Continued*

| Reagent type (species) or resource | Designation | Source or reference | Identifiers | Additional information |
|---|---|---|---|---|
| Antibody | Aggrecan Rabbit polyclonal | ABclonal | Cat# A8536 RRID:AB_632138 | IF (1:100) |
| Antibody | goat anti-Rabbit Alexa Fluor 488 | ThermoFisher Scientific | Cat# A-11008 RRID:AB_143165 | IF (1:500) |
| Antibody | ProLong Gold Antifade Mountant with DAPI | ThermoFisher Scientific | Cat# P36941 | |
| Peptide, recombinant protein | Recombinant Human sRANK Ligand | PeproTech | Cat# 310–01 | 40 ng/mL |
| Peptide, recombinant protein | Murine M-CSF | PeproTech | Cat# 315–02 | 20 ng/ml |
| Peptide, recombinant protein | Recombinant human OPG | PeproTech | Cat# 450–14 | Refer to Figure legends |
| Chemical compound, drug | prostaglandin E2 | MilliporeSigma | Cat# P0409 | 1 µM |
| Chemical compound, drug | 1α,25-Dihydroxyvitamin D3 | MilliporeSigma | Cat# D1530 | 10 nM |
| Chemical compound, drug | Collagenase | Worthington | Cat#LS004177 | 1 mg/ml |
| Chemical compound, drug | Dispase | Thermo Fisher Scientific | Cat#17105041 | 2 mg/ml |
| Commercial assay or kit | Mouse Tartrate Resistant Acid Phosphatase (TRAP) ELISA Kit | MyBioSource.com | MBS1601167 | |
| Commercial assay or kit | Mouse Osteoprotegerin ELISA Kit | MilliporeSigma | RAB0493 | |
| Commercial assay or kit | Mouse PINP ELISA Kit | MyBioSource.com | MBS2500076 | |
| Commercial assay or kit | Dual-Luciferase Reporter Assay system | Promega | E1910 | |
| Software, algorithm | Seurat | PMID:29608179 | RRID:SCR_016341 | https://satijalab.org/seurat/get_started.html |
| Software, algorithm | GraphPad Prism 8 | GraphPad Software | RRID:SCR_002798 | |
| Software, algorithm | ZEN (blue edition) version 3.4 | ZEN (blue edition) | RRID:SCR_013672 | https://www.zeiss.com/microscopy/en/products/software/zeiss-zen.html |

## Animals

*Malat1*^-/- (**Nakagawa et al., 2012**) and *Malat1*^flox/flox^ mice (**Eißmann et al., 2012**) were described previously. Sex- and age-matched *Malat1*^-/- mice and their littermates WT (*Malat1*^+/+^) mice were used for experiments. We generated mice with osteoblast-specific deletion of *Malat1* by crossing *Malat1*^flox/flox^ mice with osteocalcin cre mice (The Jackson Laboratory, Stock No: 019509). Sex- and age-matched *Malat1*^flox/flox^;*Ocn*-Cre mice (referred to as *Malat1* cKO^Ocn^) and their littermates *Malat1*^flox/flox^ mice as the controls (referred to as the *Malat1*^f/f^) were used for experiments. We generated mice with myeloid/macrophage-specific deletion of *Malat1* by crossing the *Malat1*^flox/flox^ mice with *Lyz2* Cre mice (The Jackson Laboratory, Stock No: 004781). Sex- and age-matched *Malat1*^flox/flox^;*Lyz2*-Cre mice (referred to as the *Malat1* cKO^Lyz2^ mice) and their littermates with *Malat1*^+/+^; *Lyz2* cre (+) genotype as WT controls (hereafter referred to as Control) were used for experiments.

After sacrifice, the bones were fixed by 4% formaldehyde and subjected to µCT analysis, sectioning, TRAP staining and histological analysis. µCT analysis of femoral trabecular bones and cortical midshaft was conducted to evaluate bone volume and 3D bone architecture using a Scanco µCT-35 scanner (SCANCO Medical) according to the manufacturer's instructions and the American Society of Bone and Mineral Research (ASBMR) guidelines (**Bouxsein et al., 2010**).

For dynamic histomorphometric measures of bone formation (**Deng et al., 2020**), calcein (25 mg/kg, Sigma) was injected into mice intraperitoneally at 5 and 2 days before sacrifice to obtain double

labeling of newly formed bones. The non-decalcified tibia bones were embedded in methyl meth-acrylate. 5 mm thick sections were sliced using a microtome (Leica RM2255, Leica Microsystems, Germany). For static histomorphometric measures of osteoblast parameters, non-decalcified sections of the tibiae were stained using toluidine blue or Masson-Goldner staining kit (MilliporeSigma). The Osteomeasure software was used for bone histomorphometry using standard procedures according to the program's instruction.

All mice were housed in a 12 hr:12 hr light/dark cycle with food and water ad libitum. All animal procedures were performed according to the approved protocol (2016–0001 and 0004) by the Institutional Animal Care and Use Committee (IACUC) of Hospital for Special Surgery and Weill Cornell Medical College.

## Bone fracture model

Bone fracture was performed as described previously with slight modifications (*Xu et al., 2018*). 13-week-old mice were anesthetized with 2.5% isoflurane via inhalation. The mice received meloxicam (subcutaneous injection, 2 mg/kg) and buprenorphine (subcutaneous injection, 0.5 mg/kg) as analgesia prior to surgery. The surgical site was shaved and sterilized using iodine and 70% ethanol. An incision was made over the anterolateral femur. 0.1 ml bupivacaine (5 mg/ml) was injected into the tissue adjacent to the incision line. A 25-gauge needle was inserted into the femoral canal through the patellar groove. The mid-diaphysis of the femur was transected using dental drill with a burr (19007–05, Fine science tools). The needle was then trimmed from the distal end to prevent it from projecting into the femoral joint space. The left part of the needle remained in femur to stabilize the fracture. The muscle was repositioned over the injury site and stitched using absorbable sutures. The skin was closed with wound clips, which were removed 2 weeks post-surgery. Animals were administered 0.5 mg/kg buprenorphine every 12 hours and 2.0 mg/kg meloxicam every 24 hr subcutaneously for analgesia up to 3 days after the surgery. Mice were sacrificed 21 days post-fracture. Hematoxylin and eosin (HE) staining of the bone samples were performed. BV/TV of the callus was analyzed using μCT and Osteomeasure on the tissue slices. All surgical procedures were performed according to the approved protocol (2016–0001/4) by the Institutional Animal Care and Use Committee (IACUC) of Hospital for Special Surgery and Weill Cornell Medical College.

## Reagents

Murine M-CSF (Cat# 315–02), recombinant human sRANK Ligand (Cat# 310–01), recombinant human OPG (Cat# 450–14) and murine Wnt3a (Cat# 315–20) were purchased from PeproTech. Leukocyte Acid Phosphatase (TRAP) Kit (Cat# 387 A) and Mouse Osteoprotegerin ELISA Kit (Cat# RAB0493) were obtained from MilliporeSigma. The plasmid of M50 Super 8 x TOPFlash (Cat# 12456) was purchased from Addgene. pRL-Tk control plasmid (Cat# E2241) was purchased from Promega. Fetal Bovine Serum (FBS, Cat #S11550) was obtained from Atlanta Biologicals. β-glycerophosphate disodium salt hydrate (Cat# G9422), L-ascorbic acid (Cat# A4544), prostaglandin E2 (PGE2) (Cat# P0409) and 1α,25-Dihydroxyvitamin D3 (VitD3) (Cat# D1530) were obtained from MilliporeSigma.

## Cell culture

For osteoclastogenesis, bone marrow cells were obtained from age and sex-matched WT and *Malat1*$^{-/-}$ littermates and cultured in αMEM supplemented with 10% FBS and 2.4 mM glutamine (25030081, Thermo Fisher Scientific) and 1% Penicillin–Streptomycin along with CMG14–12 supernatant (*Xia et al., 2022*), serving as the condition medium (CM) which contained a concentration equivalent to 20 ng/ml of rM-CSF and was utilized as the M-CSF source. After 3-day culture, the cells were washed in 1 x PBS and the attached cells were scraped, and seeded into plates at a density of $4.5 \times 10^4$ /cm$^2$ with CM. Next day, the cells were induced for osteoclastogenesis with 40 ng/ml RANKL and CM for the indicated days shown in figures. Multi-nucleated osteoclasts were stained using Leukocyte Acid Phosphatase (TRAP) Kit (Cat# 387 A, MilliporeSigma) according to the manufacturer's instructions.

Primary osteoblastic cells were isolated from the calvaria of new-born (0-3d) mice by enzymatic digestion in 10% FBS αMEM with 0.1% collagenase (LS004177, Worthington) and 0.2% dispase (17105041, Thermo Fisher Scientific) as described (*Deng et al., 2020*). The cells were used immediately or cultured to expand for 6 days for experiments indicated in relevant figures.

For the co-cultures of primary osteoblasts and bone marrow cells in transwell plates, primary calvarial osteoblasts isolated from WT and *Malat1*[-/-] newborn mice were seeded in the upper chamber ($2\times10^4$ cells/96-well) in αMEM medium with 10% FBS. After primary osteoblasts reached 60–70% confluence, bone marrow cells harvested from the femur and tibia of 6-week-old mice were plated to the bottom chamber ($1\times10^6$ cells/24-well), together with 1 μM Prostaglandin E2 (PGE2) (P0409, MilliporeSigma) and 10 nM 1α,25-Dihydroxyvitamin D3 (VitD3) (D1530, MilliporeSigma). Culture media were exchanged every two days for 12 days. TRAP staining was performed and multinucleated osteoclasts were counted.

For the direct co-cultures of primary osteoblasts and bone marrow cells, primary osteoblasts isolated from WT and *Malat1*[-/-] newborn mice were seeded in 48-well plates at a density of $2\times10^4$ cells/well in αMEM supplemented with 10% FBS. After osteoblasts reached 60–70% confluence, murine bone marrow cells isolated from the femur and tibia of 6-week-old mice were plated at a density of $2\times10^5$ cells/well on the top the osteoblasts in the presence of 1 μM PGE2 and 10 nM VitD3. Culture media were exchanged every two days. On the sixth day, TRAP staining was performed and multinucleated osteoclasts were counted.

Preparation and culture of primary mouse chondrocytes were performed as previously described with slight modification (*Gosset et al., 2008*). In brief, femoral condyles and tibial plateau were carefully dissected from 5-day-old WT and *Malat1*[-/-] mice. Soft tissues were removed under a microscope. After washing with PBS for two times, the femoral condyles and tibial plateau were first digested in digestion buffer (αMEM containing 10% FBS, 1 mg/ml collagenase II, and 2 mg/ml Dispase I) in 37 °C incubator for 45 min. Then, the femoral condyles and tibial plateau were transferred to fresh digestion buffer and digested overnight in 37 °C incubator. The next day, the isolated chondrocytes were filtered through a 70 μm cell strainer and spun down at 1600 rpm for 5 min. The chondrocytes were directly used for RNA extraction, seeded for Alican blue staining (A5268, MilliporeSigma) in 24-well plates, or for immunofluorescence staining of Aggrecan (A8536, ABclonal) in 96-well plates.

MC3T3-E1 osteoblasts were purchased from ATCC and cultured in αMEM with 10% FBS and 1% Penicillin–Streptomycin (15140122, Thermo Fisher Scientific). MC3T3-E1 cell line has been authenticated by STR profiling and tested negative for mycoplasma.

## Production of Wnt3a conditioned medium

The L Wnt-3A (CRL-2647, ATCC) cells and the control L cell line (CRL-2648, ATCC) were generously gifted by Dr. Joe Zhou (Weill Cornell Medicine). We followed ATCC handling information to maintain and culture the cells. Briefly, the cells were maintained in DMEM medium supplemented with 10% FBS, 1% Penicillin–Streptomycin and 0.4 mg/ml G-418. To produce the Wnt3a conditioned medium (Wnt3a CM) and control medium (L CM), the cultured cells (80–90% confluence) from one 10 cm dish were split at a 1:10 ratio and seeded into 10 cm tissue culture dishes. After an initial 4-day culture (approximately to confluency) in DMEM medium containing 10% FBS and 1% Penicillin–Streptomycin, the first batch of medium was harvested. Subsequently, 10 ml of fresh medium was added, and the cells were cultured for another 3 days to collect the second batch of conditioned medium. The two batches of conditioned media were combined at a 1:1 ratio, filtered using a 0.22 μm filter, and stored at –80 °C. Wnt3a activity was confirmed by nuclear translocation of β-catenin.

## Immunofluorescence staining

Primary osteoblasts or chondrocytes were seeded into 96-well plates at a density of $1\times10^4$ cells/well. Primary osteoblasts were serum starved for 16 hr, followed by treatment with 50% L- or 50% Wnt3a CM for 1 hr. After washing with PBS, the cells were fixed with 4% paraformaldehyde in PBS for 20 min at room temperature, permeabilized with 0.5% Triton X-100 in PBS for 15 min at room temperature, and blocked for 1 hr with 1% BSA in PBS (blocking buffer). The cells were then incubated with primary antibodies that were diluted in blocking buffer for overnight at 4 °C. After washing three times with PBS, the cells were incubated with secondary antibodies for 1 hr at room temperature. After three times of washing with PBS, the cells were mounted with the ProLong Gold Antifade Mountant with DAPI (P36941, ThermoFisher Scientific). A Zeiss microscope was used to take images. Primary antibodies include anti-β-catenin antibody (8480 s, Cell Signaling Technology, 1:100) and anti-Aggrecan (A8536, ABclonal, 1:100) in this study. The goat anti-Rabbit Alexa Fluor 488 (A-11008, ThermoFisher

Scientific, diluted at 1:500 with blocking buffer) was used as the secondary antibody for these primary antibodies.

## Immunoblot analysis

Total cellular extracts were obtained using lysis buffer containing 150 mM Tris-HCl (pH 6.8), 6% SDS, 30% glycerol, and 0.03% Bromophenol Blue, with 10% 2-Mercaptoethanol added immediately before harvesting cells. Cell lysates were fractionated on 7.5% SDS-PAGE, transferred to Immobilon-P membranes (0.45 μm, Millipore), and incubated with specific antibodies. Western Lightning Plus-ECL (PerkinElmer) was used for detection. β-catenin antibody (9562, 1:1000) and Jag1 antibody (70109, 1:1000) were obtained from Cell Signaling Technology. Nfatc1 antibody (556602, 1:1000) was obtained from BD Biosciences; Blimp1 (sc-47732, 1:1000), c-Fos (sc-52, 1:1000), OPG/Osteoprotegerin (sc-390518, 1:1000) and p38α (sc-535, 1:3000) antibodies were purchased from Santa Cruz Biotechnology.

## Cytoplasmic and nuclear extraction

The cultured primary osteoblastic cells were subjected to serum starvation in αMEM with 2% FBS for 16 hr. Subsequently, the cells were treated with either 50% L- or 50% Wnt3a conditioned medium (CM) for 1 hr. The cells were collected and lysed with Buffer A (10 mM Hepes PH7.9, 1.5 mM MgCl$_2$, 10 mM KCl) supplemented with fresh protease inhibitor cocktail (11836170001, MilliporeSigma). After incubation on ice for 15 min, 0.2 % NP-40 was added. The mixture was vortexed and incubated on ice for 2 min. The cellular lysate was then centrifuged at 10,000 × $g$ for 3 min at 4 °C. The supernatant was used as cytoplasmic fraction. The nuclei pellets were washed two times using 1 ml of Buffer A. After the second wash, the buffer was completely removed. The nuclei pellet was then lysed using Buffer C (20 mM Hepes pH7.9, 1.5 mM MgCl$_2$, 420 mM NaCl, 0.2 mM EDTA, 25% Glycerol) with fresh protease inhibitor cocktail for 30 mins on ice, vortexing every 5 min. The nuclear lysate was centrifuged at 12,000 x $g$ at 4 °C for 15 min, and the supernatant was collected as the nuclear extract. GAPDH (sc-25778) and TBP1(sc-204) were purchased from Santa Cruz Biotechnology and used as the cytoplasmic and nuclear markers, respectively. β-catenin antibody (9562, 1:1000) was obtained from Cell Signaling Technology.

## Reverse transcription and real-time PCR

DNA-free RNAs were isolated from cells with the RNeasy MiniKit (74104, QIAGEN) with DNase treatment, and total RNA was reverse-transcribed with random hexamers using the RevertAid RT Kit (K1691, Thermo Fisher Scientific) according to the manufacturer's instructions. Real-time PCR was done in triplicate with the QuantStudio 5 Real-time PCR system (A28138, Applied Biosystems) and Fast SYBR Green Master Mix (4385612, Thermo Fisher Scientific) with 500 nM primers. mRNA amounts were normalized relative to glyceraldehyde-3-phosphate dehydrogenase (GAPDH) mRNA. The mouse primers for real-time PCR were as follows: *Gapdh*: 5'-ATCAAGAAGGTGGTGAAGCA-3' and 5'-AGAC AACCTGGTCCTCAGTGT-3'; *Tnfrsf11b*: 5'-CGGAAACAGAGAAGCCACGCAA-3' and 5'-CTGTCCAC CAAAACACTCAGCC-3'; *Tnfsf11*: 5'-CAGCATCGCTCTGTTCCTGTA-3' and 5'-CTGCGTTTTCAT GGAGTCTCA-3'; *Axin2*: 5'-ATGCAAAAGCCACCCAAAGG-3' and 5'-TGCATTCCGTTTTGGCAAGG -3'; *Ccnd1*: 5'-GCGTACCCTGACACCAATCTC-3' and 5'-CTCCTCTTCGCACTTCTGCTC-3'; *Lef1*: 5'-TGTTTATCCCATCACGGGTGG-3' and 5'-CATGGAAGTGTCGCCTGACAG-3'; c-myc: 5'-CAGCGACT CTGAAGAAGAGCA-3' and 5'-TTGTGCTGGTGAGTGGAGAC-3'; *Jag1*: 5'- TGCCTGCCGAACCCCT GTCATAAT-3' and 5'- CCGATACCAGTTGTCTCCGTCCAC-3'; *Tcf7*: AACTGGCCCGCAAGGAAAG and CTCCGGGTAAGTACCGAATGC;*Nfatc1*: 5'-CCCGTCACATTCTGGTCCAT-3' and 5'-CAAGTAAC CGTGTAGCTCCACAA-3'; *Acp5*: 5'-ACGGCTACTTGCGGTTTC-3' and 5'-TCCTTGGGAGGCTGGT C-3'; *Ctsk:* 5'-AAGATATTGGTGGCTTTGG-3' and 5'-ATCGCTGCGTCCCTCT-3'; *Dc-stamp*: 5'-TTTG CCGCTGTGGACTATCTGC-3' and 5'-AGACGTGGTTTAGGAATGCAGCTC-3'; *Blimp1*: 5'-TTCTTGTG TGGTATTGTCGGGACTT-3' and 5'-TTGGGGACACTCTTTGGGTAGAGTT-3'; *Atp6v0d2*: 5'-GAAG CTGTCAACATTGCAGA-3' and 5'-TCACCGTGATCCTTGCAGAAT-3'; *Malat1*: 5'-AGCAGGCATTGT GGAGAGGA-3' and 5'-ATGTTGCCGACCTCAAGGAA-3'; *Col2a1*: CGATCACAGAAGACCTCCCG and GCGGTTGGAAAGTGTTTGGG; *Sox9:* AAGCTCTGGAGGCTGCTGAACGAG and CGGCCTCC GCTTGTCCGTTCT; *Acan*: GGTCACTGTTACCGCCACTT and CCCCTTCGATAGTCCTGTCA.

## RNA immunoprecipitation (RIP) assay

Thirty million MC3T3-E1 cells were collected, centrifuged, and washed with PBS. The cells were then spun down at 800 x $g$ for 4 min at room temperature. The cell pellet was resuspended in 1% formaldehyde in PBS (28906, Thermo Fisher Scientific) and crosslinked for 10 min at room temperature on an end-to-end rotator. 1.25 M glycine at 1/10 volume of 1% formaldehyde solution was used to quench the cross-linking reaction at room temperature for 5 min. The cells were spun down at 2000 x $g$ for 5 min and the pellet was washed with chilled PBS once, followed by centrifugation at 2000 x $g$ for 5 min. The cell pellets were resuspended in 1.1 ml immunoprecipitation (IP) lysis buffer (50 mM HEPES at pH 7.5, 0.4 M NaCl, 1 mM EDTA, 1 mM DTT, 0.5% Triton X-100, 10% glycerol) containing 1 mM PMSF (78830, MilliporeSigma), protease inhibitor cocktail, and RNase inhibitor (100 U/ml Superase-in, AM2694, Thermo Fisher Scientific). The cell lysate was then sonicated using the Bioruptor Pico sonication device (Diagenode, NJ, USA) until the liquid became clear (sonication cycle: 30 s ON, 30 s OFF). The lysates were centrifuged for 10 min at 14,000 x $g$ at room temperature. 50 μl of the supernatant was used as the input control. The remaining supernatant was precleared using protein A/G agarose (sc-2003, Santa Cruz Biotechnology). The precleared cell lysate was split into two tubes evenly and incubated with 5 μg of β-catenin antibody (9562, Cell signaling technology) or normal rabbit IgG (2729 S, Cell signaling technology) at 4 °C overnight with rotation. 30 μl of washed protein A/G beads were then added into each sample and incubated at 4 °C for 1 h with rotation. The beads were washed with 900 μl of IP lysis buffer for 3 min/each time for five times and collected by centrifugation for 3 min at 400 x $g$ at room temperature. After the last wash, 100 μl of RIP buffer (50 mM HEPES at pH 7.5, 0.1 M NaCl, 5 mM EDTA, 10 mM DTT, 0.5% Triton X-100, 10% glycerol, 1% SDS) with 1 μl RNase inhibitor was added to each sample. 50 μl of RIP buffer was added to the input control sample. All samples were incubated at 70 °C for 1 hr to reverse crosslinking. 100 μl of supernatant were then collected by spinning down the beads at 400 x $g$ for 1 min at room temperature. The supernatant was used for RNA extraction using the RNeasy Mini Kit (QIAGEN) with DNase I treatment. RNA was eluted using 12 μl RNase-free H2O and reversed to complementary DNA using One-step cDNA synthesis kit (Thermo Fisher, Revert Aid RT kit, k1691). The qPCR was then performed using *Malat1* primers.

## Chromatin isolation by RNA purification (ChIRP) assay

Chromatin Isolation by RNA Purification (ChIRP) was performed as described previously with slight modifications (*Chu et al., 2012*). We designed and synthesized an antisense oligonucleotide probe of murine *Malat1*. The design was based on the high sequence homology to a human *MALAT1* probe's sequence (*West et al., 2014*; *Figure 3—figure supplement 1*). The murine *Malat1* probe's sequence is 5'-GTCTTTCCTGCCTTAAAGTTAATTTCG/iSp18//3'-BiotinTEG (INTEGRATED DNA Technologies). We also synthesized a GFP probe (sequence: 5' TATCACCTTCAAACTTGACTTC/iSp18-3'-Biotin TEG) as the negative control. 30 million MC3T3-E1 cells were collected, washed in PBS and centrifuged at 800xg for 4 min at room temperature. The cell pellet was resuspended in 4% formaldehyde in PBS and crosslinked for 30 min at room temperature on an end-to-end rotator. 1.25 M glycine at 1/10 volume of 4% formaldehyde solution was used to quench the crosslinking reaction at room temperature for 5 min. After washing with chilled PBS once, the cells were resuspended in 1 ml lysis buffer (50 mM Tris-Cl pH 7.0, 10 mM EDTA, 1% SDS) supplement with 1 mM PMSF, protease inhibitors, and RNase inhibitor. The cell lysate was sonicated using the Bioruptor until the lysate became clear (sonication cycle: 30 s ON, 30 s OFF). The lysate was centrifuged for 10 min at 14,000 x $g$ at 4 °C. The sonicated lysate was then precleared with magnetic streptavidin beads (65001, Thermo Fisher Scientific) at 37 °C for 30 min with slow rotation. 2% volume of precleared lysate was saved for the input. The remaining lysate was split into two new tubes evenly and incubated with 100 pmol of the *Malat1* probe or the negative control probe in the hybridization buffer (750 mM NaCl, 1% SDS, 50 mM Tris-Cl pH 7.0, 1 mM EDTA, 15% formamide containing 1 mM PMSF, protease inhibitor cocktail (11836170001, MilliporeSigma), and RNase inhibitor (100 U/ml Superase-in, AM2694, Thermo Fisher Scientific)) at 37 °C for overnight with slow rotation. Magnetic streptavidin beads were then added to the samples and incubated at 37 °C for 30 min with slow rotation. DynaMag-15 magnetic strip was used to separate beads from the liquids. After 5 times of washing using the wash buffer (2 x NaCl and Sodium citrate (SSC, 15557044, Thermo Fisher Scientific), 0.5% SDS) with PMSF and proteinase inhibitor cocktail, the beads were isolated. The bound proteins were eluted by boiling the beads in 30 μl of 3 x blue juice buffer (150 mM Tris-HCl (pH

6.8), 6% SDS, 30% glycerol, and 0.03% Bromophenol Blue, with fresh 10% 2-Mercaptoethanol) and subjected to immunoblot analysis.

## Luciferase reporter assay

$3.5 \times 10^4$ of primary calvarial osteoblasts were plated in 48-well plates. Next day, 480 ng of the M50 Super 8 x TOPFlash plasmid (12456, Addgene) and 20 ng of pRL-Tk control plasmid (E2241, Promega) were transfected per well using Lipofectamine 3000 (L3000001, Invitrogen) according to the manufacturer's instructions. After 24 hr transfection, the culture medium was replaced with fresh αMEM medium supplemented with 10%FBS. 48 hr post transfection, the cells were serum starved for 1 hr and then treated with 20% L- or 20% Wnt3a CM for 16 hr. Firefly and Renilla luciferase activities were measured using a Dual-Luciferase Reporter Assay system (E1910, Promega) on a Gen5 Microplate Reader (BioTek) according to the manufacturer's instructions. Firefly luciferase activity was normalized to Renilla luciferase activity.

## Single cell RNAseq (scRNAseq) analysis

Single cell RNAseq datasets for bone (GSM3674239, GSM3674240, GSM3674241, GSM3674242) and bone marrow (GSM3674243, GSM3674244, GSM3674246) were downloaded from GSE128423 (*Baryawno et al., 2019*). Seurat package (*Stuart et al., 2019*) was applied for downstream analysis. Briefly, genes expressed in fewer than 3 cells and cells with less than 500 genes were filtered out. Cells with over 10% mitochondrial reads and exceeding 6000 nFeature_RNA were excluded. After NormalizeData, the top 5000 variable genes were selected based on dispersion method using FindVariableGenes function of Seurat package. Subsequently, data scaling was performed using the ScaleData function. All datasets were integrated based on the identified anchors. The first 30 principal components for both UMAP (Uniform Manifold Approximation and Projection) and the subsequent application of a graph-based clustering approach were used, with resolution at 0.1. The Clustree function was executed to understand how the structure of clusters changes across different resolutions. The FindAllMarkers function was utilized with parameters set to prioritizing positive markers expressed in at least 10% of cells within a cluster and exhibiting a log-fold change threshold of 0.25. The following gene markers were also included for cluster annotation: *Acan, Col2a1 and Sox9* for chondrocytes *Baryawno et al., 2019*; *Acta2, Myh11* and *Mcam* for pericytes *Baryawno et al., 2019*; *Adipoq, Lepr, Cxcl12, Cebpa, Kitl,* and *Lpl* for Adipoq-lineage progenitors *Inoue et al., 2023*; *Prrx1, Col1a1, Ibsp,* and *Bglap* for osteoblast lineage *Inoue et al., 2023*; *Pecam1* and *Cdh5* for endothelial cells *Inoue et al., 2023*; *Cd19* for B cells *Inoue et al., 2023*; *S100a8* for neutrophils *Inoue et al., 2023*; *Mpz, Mbp,* and *Plp1* for Schwann cells *Direder et al., 2022*; *Gypa, Alas2, Snca, Hbb-bs, Hbb-bt, Car1, Car2, Klf1, Gata1,* and *Gata2* for Erythroid cells *Herkt et al., 2018*; *Locascio et al., 2015*; *Jain et al., 2022*; *Paul et al., 2015*; *Pf4, Itga2b,* and *Fli1* for Megakaryocytes *Paul et al., 2015*; *Cd68, Lyz2, Ly6c2,* and *Sell* for Monocyte-Macrophage lineage *Inoue et al., 2023*; *Ito et al., 2021*. We utilized Seurat's FeaturePlot to visualize gene expression in individual cells. DotPlot was employed to illustrate the percentage of cells in a cluster expressing a specific gene and to visualize the average scaled expression level of each gene. The distribution of normalized gene expression levels across all cells in each cluster was visualized using VlnPlot function in Seurat. R version 4.3.2 and Seurat 5.0.1 were used in the study.

## Bone marrow supernatant collection

Bone marrow from 12-week-old WT and *Malat1*[-/-] littermate mice was harvested from the femur and tibia (with both ends cut open) by flash centrifugation. The cell pellets were resuspended in 0.2 ml of chilled PBS containing protease inhibitor cocktail (11836170001, MilliporeSigma) and incubated on ice for 30 min. The suspension was centrifuged at 1500 rpm for 15 min at 4 °C and the supernatant was collected for immunoblot analysis.

## ELISA

Mouse serum OPG, P1NP and TRAP were measured using Mouse Osteoprotegerin ELISA Kit (MilliporeSigma), P1NP and TRAP ELISA kits (MyBioSource), respectively, according to the manufacturer's instruction.

## Statistical analysis

Statistical analysis was performed using Graphpad Prism software. Two-tailed Student's t test was applied when there were only two groups of samples. In the case of more than two groups of samples,

one-way ANOVA will be used with one condition, and two-way ANOVA was used with more than one condition. ANOVA analysis was followed by post hoc Bonferroni's correction for multiple comparisons. $p < 0.05$ was taken as statistically significant. Data are presented as the mean $\pm$ SD as indicated in the figure legends.

## Acknowledgements

We thank Ziyu Chen for mouse breeding and technical assistance, Matthew B Greenblatt for sharing microCT scanner. We are grateful to the lab members from Dr. Baohong Zhao's laboratory for their helpful discussions and assistance, and Peyton Carpen for reading the manuscript and valuable discussion. This work was supported by grants from the National Institutes of Health (AR071463 and AR078212 to BZ) and by support for the Rosensweig Genomics Center at the Hospital for Special Surgery from The Tow Foundation. KVP laboratory is supported by grants from NIGMS/NIH (GM132458) and NIA/NIH (AG065748). The content of this manuscript is solely the responsibilities of the authors and does not necessarily represent the official views of the NIH.

## Additional information

### Competing interests

Baohong Zhao: Reviewing editor, eLife. The other authors declare that no competing interests exist.

### Funding

| Funder | Grant reference number | Author |
|---|---|---|
| National Institutes of Health | AR071463 | Baohong Zhao |
| National Institutes of Health | AR078212 | Baohong Zhao |
| National Institutes of Health | GM132458 | Kannanganattu V Prasanth |
| National Institutes of Health | AG065748 | Kannanganattu V Prasanth |

The funders had no role in study design, data collection and interpretation, or the decision to submit the work for publication.

### Author contributions

Yongli Qin, Data curation, Software, Formal analysis, Validation, Investigation, Visualization, Methodology, Writing – original draft, Project administration, Writing – review and editing; Jumpei Shirakawa, Data curation, Formal analysis, Validation, Investigation, Visualization, Methodology, Writing – original draft, Writing – review and editing; Cheng Xu, Investigation, Visualization, Methodology, Writing – review and editing; Ruge Chen, Investigation, Visualization, Methodology, Writing – review and editing, scRNAseq analysis; Xu Yang, Supervision, Visualization, Methodology; Courtney Ng, Shinichi Nakano, Investigation, Visualization; Mahmoud Elguindy, Investigation; Zhonghao Deng, Investigation, Visualization, Writing – review and editing; Kannanganattu V Prasanth, Resources, Visualization, Writing – review and editing; Moritz F Eissmann, Shinichi Nakagawa, Resources, Visualization; William M Ricci, Supervision, Visualization, Methodology, Writing – review and editing; Baohong Zhao, Conceptualization, Resources, Supervision, Funding acquisition, Visualization, Writing – original draft, Writing – review and editing

### Author ORCIDs

Yongli Qin ⬡ https://orcid.org/0000-0001-6133-0511
Shinichi Nakagawa ⬡ https://orcid.org/0000-0002-6806-7493
Baohong Zhao ⬡ https://orcid.org/0000-0002-1286-0919

## Ethics

All animal procedures were performed according to the approved protocol (2016-0001 and 0004) by the Institutional Animal Care and Use Committee (IACUC) of Hospital for Special Surgery and Weill Cornell Medical College.

Reviewer #2 (Public review): https://doi.org/10.7554/eLife.98900.3.sa1
Reviewer #3 (Public review): https://doi.org/10.7554/eLife.98900.3.sa2
Author response https://doi.org/10.7554/eLife.98900.3.sa3

---

## Additional files

### Supplementary files
• MDAR checklist

### Data availability

All data supporting the findings of this study are available within the paper, its Supplementary Information, and source data file. The sequence dataset from GSE128423 was reanalyzed. Customized computational scripts of analyzing OPG data were deposited in Zenodo (https://doi.org/10.5281/zenodo.10421692) and GitHub (https://github.com/RugeC/OPG-R-script, copy archived at *Chen, 2024*).

The following dataset was generated:

| Author(s) | Year | Dataset title | Dataset URL | Database and Identifier |
|---|---|---|---|---|
| Chen R | 2023 | OPG R script (v1.0) | https://doi.org/10.5281/zenodo.10421692 | Zenodo, 10.5281/zenodo.10421692 |

The following previously published dataset was used:

| Author(s) | Year | Dataset title | Dataset URL | Database and Identifier |
|---|---|---|---|---|
| Regev A, Scadden D | 2019 | A cellular taxonomy of the bone marrow stroma in homeostasis and leukemia demonstrates cancer-crosstalk with stroma to impair normal tissue function | https://www.ncbi.nlm.nih.gov/geo/query/acc.cgi?acc=GSE128423 | NCBI Gene Expression Omnibus, GSE128423 |

---

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
