## [Editor Report · eLife Assessment]

This is an **important** and **convincing** dataset shedding new light on a role for Malat1 in osteoblast physiology. The work is of value to areas other than the bone field because it supports a role and mechanism for beta-catenin that is novel and unusual. The findings are significant in that they support the presence of another anabolic pathway in bone that can be productively targeted for therapeutic goals. Revisions further improved the paper and addressed the reviewers' concerns.

---

## [Referee Report · Reviewer #2 (Public review)]

Summary:

The authors investigated the roles of IncRNA Malat1 in bone homeostasis which was initially believed to be non-functional for physiology. They found that both Malat1 KO and conditional KO in osteoblast lineage exhibit significant osteoporosis due to decreased osteoblast bone formation and increased osteoclast resorption. More interestingly, they found that deletion of Matat1 in osteoclast lineage cell does not affect osteoclast differentiation and function. Mechanistically, they found that Malat1 acts as an co-activator of b-Catenin directly regulating osteoblast activity and indirectly regulating osteoclast activity via mediating OPG, but not RANKL expression in osteoblast and chondrocyte. Their discoveries establish a previous unrecognized paradigm model of Malat1 function in the skeletal system, providing novel mechanistic insights into how a lncRNA integrates cellular crosstalk and molecular networks to fine tune tissue homeostasis, remodeling.

Strengths:

The authors generated global and conditional KO mice in osteoblast and osteoclast lineage cells and carefully analyzed the role of Matat1 with both in vivo and in vitro system. The conclusion of this paper is mostly well supported by data.

Comments on revised version:

The authors have addressed all my concerns.

---

## [Referee Report · Reviewer #3 (Public review)]

Summary:

In this manuscript, Qin and colleagues study the role of Malat1 in bone biology. This topic is interesting given the role of lncRNAs in multiple physiologic processes. A previous study (PMID 38493144) suggested a role for Malat1 in osteoclast maturation. However, the role of this lncRNA in osteoblast biology was previously not explored. Here, the authors note osteopenia with increased bone resorption in mice lacking Malat1 globally and in osteoblast lineage cells. At the mechanistic level, the authors suggest that Malat1 controls beta-catenin activity. These result advance the field regarding the role of this lncRNA in bone biology.

Strengths:

The manuscript is well-written and data are presented in a clear and easily understandable manner. The bone phenotype of osteoblast-specific Malat1 knockout mice is of high interest. The role of Malat1 in controlling beta-catenin activity and OPG expression is interesting and novel.

Weaknesses:

The lack of a bone phenotype when Malat1 is deleted with LysM-Cre is of interest given the previous report suggesting a role for this lncRNA in osteoclasts, especially in light of satisfactory deletion efficiency in this model. The data in the fracture model in Figure 8 is enhanced with quantitative data. The role of Malat1 and OPG in chondrocytes is unclear since the osteocalcin-Cre mice (which should retain normal Malat1 levels in chondrocytes) have similar bone loss as the global mutants.

Comments on revised version:

All previous comments have been addressed in a satisfactory manner.

---

## [Author Response]

The following is the authors’ response to the original reviews.

**eLife Assessment**
The work is important and of potential value to areas other than the bone field because it supports a role and mechanism for beta-catenin that is novel and unusual. The findings are significant in that they support the presence of another anabolic pathway in bone that can be productively targeted for therapeutic goals. The data for the most part are convincing. The work could be strengthened by better characterizing the osteoclast KO of Malat1 related to the Lys cre model and by including biochemical markers of bone turnover from the mice.

We thank the editors and reviewers for their time and their positive and insightful comments. We are pleased that the editors and reviewers were very enthusiastic, as stated in their Strength comments. We have performed experiments and addressed all of the points raised by the reviewers. We have revised the manuscript accordingly and the reviewers’ points are specifically addressed below.

**Public Reviews:**

**Reviewer #1 (Public Review):**
SummaryThe authors were trying to discover a novel bone remodeling network system. They found that an IncRNA Malat1 plays a central role in the remodeling by binding to β-catenin and functioning through the β-catenin-OPG/Jagged1 pathway in osteoblasts and chondrocytes. In addition, Malat1 significantly promotes bone regeneration in fracture healing in vivo. Their findings suggest a new concept of Malat1 function in the skeletal system. One significantly different finding between this manuscript and the competing paper pertains to the role of Malat1 in osteoclast lineage, specifically, whether Malat1 functions intrinsically in osteoclast lineage or not.Strengths:This study provides strong genetic evidence demonstrating that Malat1 acts intrinsically in osteoblasts while suppressing osteoclastogenesis in a non-autonomous manner, whereas the other group did not utilize relevant conditional knockout mice. As shown in the results, Malat1 knockout mouse exhibited abnormal bone remodeling and turnover. Furthermore, they elucidated molecular function of Malat1, which is sufficient to understand the phenotype in vivo.We are grateful to the reviewer for highlighting the novelty, strengths and significance of our work.Weaknesses:Discussing differences between previous paper and their status would be highly informative and beneficial for the field, as it would elucidate the solid underlying mechanisms.

These points have been fully addressed in the point-to-point response below.

**Reviewer #2 (Public Review):**
Summary:The authors investigated the roles of IncRNA Malat1 in bone homeostasis which was initially believed to be non-functional for physiology. They found that both Malat1 KO and conditional KO in osteoblast lineage exhibit significant osteoporosis due to decreased osteoblast bone formation and increased osteoclast resorption. More interestingly they found that deletion of Malat1 in osteoclast lineage cells does not affect osteoclast differentiation and function. Mechanistically, they found that Malat1 acts as a co-activator of b-Catenin directly regulating osteoblast activity and indirectly regulating osteoclast activity via mediating OPG, but not RANKL expression in osteoblast and chondrocyte. Their discoveries establish a previously unrecognized paradigm model of Malat1 function in the skeletal system, providing novel mechanistic insights into how a lncRNA integrates cellular crosstalk and molecular networks to fine-tune tissue homeostasis, and remodeling.Strengths:The authors generated global and conditional KO mice in osteoblast and osteoclast lineage cells and carefully analyzed the role of Matat1 with both in vivo and in vitro systems. The conclusion of this paper is mostly well supported by data.

We are grateful to the reviewer for highlighting the novelty, strengths and significance of our work.

Weaknesses:More objective biological and biochemical analyses are required.

These points have been fully addressed in the point-to-point response below.

**Reviewer #3 (Public Review):**
Summary:In this manuscript, Qin and colleagues study the role of Malat1 in bone biology. This topic is interesting given the role of lncRNAs in multiple physiologic processes. A previous study (PMID 38493144) suggested a role for Malat1 in osteoclast maturation. However, the role of this lncRNA in osteoblast biology was previously not explored. Here, the authors note osteopenia with increased bone resorption in mice lacking Malat1 globally and in osteoblast lineage cells. At the mechanistic level, the authors suggest that Malat1 controls beta-catenin activity. These results advance the field regarding the role of this lncRNA in bone biology.Strengths:The manuscript is well-written and data are presented in a clear and easily understandable manner. The bone phenotype of osteoblast-specific Malat1 knockout mice is of high interest. The role of Malat1 in controlling beta-catenin activity and OPG expression is interesting and novel.

We are grateful to the reviewer for highlighting the novelty, strengths and significance of our work.

Weaknesses:The lack of a bone phenotype when Malat1 is deleted with LysM-Cre is of interest given the previous report suggesting a role for this lncRNA in osteoclasts. However, to interpret the findings here, the authors should investigate the deletion efficiency of Malat1 in osteoclast lineage cells in their model. The data in the fracture model in Figure 8 seems incomplete in the absence of a more complete characterization of callus histology and a thorough time course. The role of Malat1 and OPG in chondrocytes is unclear since the osteocalcin-Cre mice (which should retain normal Malat1 levels in chondrocytes) have similar bone loss as the global mutants.

These points have been fully addressed in the point-to-point response below.

**Recommendations for the authors:**

**Reviewing Editor (Recommendations For The Authors):**
There are several suggestions for improving the manuscript, and we hope that you will review the recommendations carefully and make changes to the paper to address the concerns raised. Suggestions have been made to better characterize the osteoclast KO of Malat1 related to the Lys cre model as well as suggestions to include biochemical markers of bone turnover from your mice.

These points have been fully addressed in the point-to-point response below.

**Reviewer #1 (Recommendations For The Authors):**
(1) Replicate numbers in Figure 3 should be noted.

We thank the reviewer for this point. The experiments in Fig. 3 have been replicated three times, which is now noted in the figure legend.

(2) It is novel to identify OPG expression in chondrocytes. More discussion is expected.

Yes, a paragraph regarding this point has been added to the Discussion section.

**Reviewer #2 (Recommendations For The Authors):**
(1) It is better to show serum osteoblast bone formation marker and osteoclast resorption marker, such as P1NP and CTx, in both Malat1 KO and osteoblast conditional KO mice.

We thank the reviewer for this important point. Since CTx values are often influenced by food intake, we measured serum TRAP levels, which also reflect changes in osteoclastic bone resorption. We have observed that the serum osteoblastic bone formation marker P1NP was decreased, while osteoclastic bone resorption marker TRAP was increased, in both *Malat1^-/-^* and *Malat1^ΔOcn^* mice. These changes in serum biochemical markers of bone turnover are consistent with the bone phenotype caused by Malat1 deficiency. The new data are shown in Fig.1i, Fig. 2e, and Fig.5b.

(2) in vitro osteoblast differentiation assay is required to further confirm Malat1 regulates osteoblast differentiation.

We thank the reviewer for this suggestion. As recommended, we have performed in vitro osteoblast differentiation multiple times using calvarial cells, a commonly used system in the field. However, we observed big variability in the culture results across different experimental batches, whether conducted by different scientists or the same individual. This variability is likely due to differences in the purity of the cultured cells, as literature shows that the current culture system in the field contains a mixture of tissue cells, including not only osteoblasts but also other cells, such as stromal and hematopoietic lineage cells (DOI: 10.1002/jbmr.4052). We hope to test osteoblast differentiation using a purer culture system once it becomes available in the field. In contrast, our in vivo data, indicated by multiple parameters, show consistent osteoblast and bone formation phenotypes across a large number of mice. Therefore, the in vivo results in our study strongly support our conclusion regarding Malat1's role in osteoblastic bone formation.

(3) The authors found that Matat1 regulates osteoclast activity through OPG expression not only in osteoblasts, but also in chondrocytes and concluded that chondrocyte is involved in the crosstalk with osteoclast lineage cells in marrow. This is a very novel finding. Do the authors have any in vivo data to support this point, such as deleting Malat1 in chondrocyte lineage cells with chondrocyte-specific Cre?

We appreciate the reviewer for highlighting our novel findings and providing valuable suggestions. Given the considerable time required to generate chondrocyte-specific conditional KO mice, we plan to thoroughly investigate the crosstalk between chondrocytes and osteoclasts via Malat1 in vivo in our next project.

**Reviewer #3 (Recommendations For The Authors):**
(1) Ideally would show male and female data side by side in the main text figures

We thank the reviewer for this suggestion. The male and female data are now displayed side by side in Fig. 1b.

(2) The sample size for the in vivo datasets is quite large. A power calculation should be provided to better understand how the authors decided to analyze so many mice.

Due to staff turnover during the pandemic, the first authors and several co-authors were involved in breeding the mice and collecting and analyzing bone samples. To avoid bias in sample selection, we pooled all the samples, resulting in a highly consistent phenotype across mice. This robust approach further strengthens our conclusion.

(3) The candidate gene approach to look at beta-catenin is a bit random, it would be ideal to assess Malat1 binding proteins in osteoblasts in an unbiased way. Also, does Malat1 bind bcatenin in other cell types? The importance of this point is further underscored by ref 47 which indicates that Malat binds TEAD3.

As β-catenin is a key regulator in osteoblasts, we believe that studying the interaction between β-catenin and Malat1 is not random. Instead, this approach is well-founded and based on established knowledge in the field (as discussed below). In parallel, we are investigating genome-wide Malat1-bound targets beyond β-catenin, which will be reported in future studies.

More detailed points have been discussed in the manuscript:

Given that we identified Malat1 as a critical regulator in osteoblasts, we sought to investigate the mechanisms underlying the regulation of osteoblastic bone formation by Malat1. β-catenin is a central transcriptional factor in canonical Wnt signaling pathway, and plays an important role in positively regulating osteoblast differentiation and function (28-33). Upon stimulation, most notably from canonical Wnt ligands, β-catenin is stabilized and translocates into the nucleus, where it interacts with coactivators to activate target gene transcription. Previous reports observed a link between Malat1 and β-catenin signaling pathway in cancers (34,35), but the underlying molecular mechanisms in terms of how Malat1 interacts with β-catenin and regulates its nuclear retention and transcriptional activity are unclear.

Ref47 tested Malat1 binding to Tead3 in osteoclasts. However, a key difference between our findings and those of Ref47 is that both our in vitro and in vivo data, using myeloid osteoclastspecific conditional Malat1 KO mice, do not support an intrinsically significant role for Malat1 in osteoclasts.

(4) The statement on page 6 concluding that Malat acts as a scaffold to tether β-catenin in the nucleus is not supported by data in Fig 3d demonstrating that b-catenin nucleus translocation in response to Wnt3a is similar in control and Malat-deficient cells.

The experiment in Fig. 3d is not designed to demonstrate Malat1 and β-catenin binding, but it is essential as the result rules out the possibility that Malat1 may affect β-catenin nuclear translocation. Moreover, we have utilized two robust approaches, CHIRP and RIP, to demonstrate that Malat1 acts as a scaffold to tether β-catenin in the nucleus (Fig. 3a, b, c, Supplementary Fig. 3).

(5) Figure 4e: can the authors show Malat deletion efficiency in the LysM-Cre model? This is important in light of the negative data in this figure and ref 47 which claims an osteoclast intrinsic role for Malat

We thank the reviewer for this suggestion. The deletion efficiency of Malat1 in the LysM-Cre mice is very high (>90%). This data is now presented in Fig. 4e.

(6) Figure 5: since the magnitude of the effects on osteoclasts at the histology level are mild, it would be nice to also look at serum markers of bone resorption (CTX)

The magnitude of osteoclast changes at the histological level in Fig. 5 is not mild in our view, as we observe 25-30% changes with statistical significance in the osteoclast parameters of *Malat1ΔOcn* mice. Since CTx values are often influenced by food intake, we measured serum TRAP levels, which reflect changes in osteoclastic bone resorption. As shown in Fig.5b, serum TRAP levels are significantly elevated in *Malat1^ΔOcn^* mice compared to control mice.

(7) Data showing chondrocytic expression of OPG is not as novel as the authors claim. Should think about growth plate versus articular sources of OPG. Growth plate chondrocytes express OPG to regulate osteoclasts in the primary spongiosa which resorb mineralized cartilage.

In the present study, we do not focus on comparing the sources of OPG from the chondrocytes in the growth plate versus articular cartilage. The novelty of our work lies in the discovery that Malat1 links chondrocyte and osteoclast activities through the β-catenin-OPG/Jagged1 axis. This Malat1-β-catenin-OPG/Jagged1 axis represents a novel mechanism regulating the crosstalk between chondrocytes and osteoclasts.

(8) The relevance of the chondrocyte role of Malat is unclear since the bone phenotype in global and osteocalcin-Cre mice is similar.

Bone mass was decreased by 20% in *Malat1^ΔOcn^* mice, while a 30% reduction was observed in global KO (*Malat1^-/-^*) mice. This difference indicates potential contributions from other cell types, such as chondrocytes, and our results in Fig. 6 further support the impact of chondrocytes in Malat1's regulation of bone mass. We plan to thoroughly investigate the crosstalk between chondrocytes and osteoclasts via Malat1 in vivo in our next project.

(9) Fracture data in Figure 8 seems incomplete, it would be ideal to support micro CT with histology and look at multiple time points.

We thank the reviewer for this suggestion. We have performed histological analysis of our samples, and found that Malat1 promotes bone healing in the fracture model (Fig. 8f), which is consistent with our μCT data.